# A flexible artificial chemosensory neuronal synapse based on chemoreceptive ionogel-gated electrochemical transistor

Hamna Haq Chouhdry [1,2], Dong Hyun Lee[3], Atanu Bag [3,4] ✉ & Nae-Eung Lee [1,2,3,4,5,6,7] ✉

The human olfactory system comprises olfactory receptor neurons, projection neurons, and interneurons that perform remarkably sophisticated functions, including sensing, filtration, memorization, and forgetting of chemical stimuli for perception. Developing an artificial olfactory system that can mimic these functions has proved to be challenging. Herein, inspired by the neuronal network inside the glomerulus of the olfactory bulb, we present an artificial chemosensory neuronal synapse that can sense chemical stimuli and mimic the functions of excitatory and inhibitory neurotransmitter release in the synapses between olfactory receptor neurons, projection neurons, and interneurons. The proposed device is based on a flexible organic electrochemical transistor gated by the potential generated by the interaction of gas molecules with ions in a chemoreceptive ionogel. The combined use of a chemoreceptive ionogel and an organic semiconductor channel allows for a long retentive memory in response to chemical stimuli. Long-term memorization of the excitatory chemical stimulus can be also erased by applying an inhibitory electrical stimulus due to ion dynamics in the chemoresponsive ionogel gate electrolyte. Applying a simple device design, we were able to mimic the excitatory and inhibitory synaptic functions of chemical synapses in the olfactory system, which can further advance the development of artificial neuronal systems for biomimetic chemosensory applications.

Signal transduction and perception in the sensory nervous system (SNS) are carried out by a series of synaptic events beginning at the sensory organ and ending in the brain[1,2]. Sensory organs with receptors convert a specific type of stimulus into action potentials and transmit pre-processed sensory signals to the central nervous system. Sensory organs, including those in tactile, auditory, visionary, gustatory, and olfactory systems, are often composed of sensory neurons with receptors at their own terminal or receptor cells innervated with afferent neurons, with the generated signals communicated by neurotransmitters across synapses to other neurons closer to the brain. Chemoreceptors, mechanoreceptors, photoreceptors, nociceptors, and thermoreceptors are the five major receptors in sensory organs

[1]SKKU Advanced Institute of Nano Technology (SAINT), Sungkyunkwan University, Suwon, Gyeonggi-do 16419, Republic of Korea. [2]Department of Nano Science and Technology, Sungkyunkwan University, Suwon, Gyeonggi-do 16419, Republic of Korea. [3]School of Advanced Materials Science & Engineering, Sungkyunkwan University, Suwon, Gyeonggi-do 16419, Republic of Korea. [4]Research Centre for Advanced Materials Technology, Sungkyunkwan University, Suwon, Gyeonggi-do 16419, Republic of Korea. [5]Samsung Advanced Institute for Health Sciences & Technology (SAIHST), Sungkyunkwan University, Suwon, Gyeonggi-do 16419, Republic of Korea. [6]Institute of Quantum Biophysics (IQB), Sungkyunkwan University, Suwon, Gyeonggi-do 16419, Republic of Korea. [7]Biomedical Institute for Convergence at SKKU (BICS), Sungkyunkwan University, Suwon, Gyeonggi-do 16419, Republic of Korea. ✉e-mail: abag@skku.edu; nelee@skku.edu

that respond to stimuli and carry out transduction[3–10]. Energy-efficient and intelligent signal processing by the SNS has sparked extensive research efforts to mimic biological sensory neuronal systems, including receptors[11–13]. Emulating the synaptic connection between sensory receptors and afferent neurons has been of great interest for researchers involved with neuromorphic sensor engineering[14–19].

Artificial sensory systems have advanced towards intelligent sensor systems that sense, filter, compute, and memorize at the device level[20–22]. Most relevant research to date has been limited to mechanoreceptors[15] or photoreceptors[22,23]. Although a few attempts have been made to artificially mimic chemoreceptive sensory systems, the development of artificial chemoreceptors has been challenging due to their complexity[24–28]. Mimicking a chemoreceptive sensory system often requires connecting chemical sensors to separate synaptic devices, making for complex structures and fabrication processes, and involves applying the concept of synaptic plasticity modulation through electrical pulsing under exposure to chemical gas, rather than chemical pulsing[29,30]. Chemoreceptors are found exclusively in olfactory and gustatory systems. Stimuli in the form of odors or tastants binding to receptor cells generate action potentials and release neurotransmitters from the receptor cells to afferent neurons. The olfactory system consists of a network of chemical synapses in the glomerulus of the olfactory bulb[31]. Synaptic events in these synapses can be excitatory or inhibitory. The olfactory bulb in particular has more inhibitory synapses compared with the other areas of the brain, which are believed to be helpful in the adaptation, perception, discrimination, regulation, and attenuation of odor signals, and odor coding (e.g., shaping the overall output of the olfactory bulb)[32–34]. The adaptation, potentiation, or inhibition of signals in chemosensory systems can occur locally by neuromodulation of excitatory and inhibitory synapses, altering conducting states and potentials[31,35]. However, no previous reports address the functions of adaptation, potentiation, or inhibition of signal in artificial olfactory systems[24,25,28,36].

Here, we present a flexible artificial chemosensory neuronal synapse (ACNS) with synaptic functions that emulate the inhibition and potentiation of biological synapses in the glomerulus of the olfactory bulb. Our ACNS is based on an organic electrochemical transistor (OECT) with a poly(3,4-ethylenedioxythiophene) polystyrene sulfonate (PEDOT:PSS) channel and an ionogel (IG) based on 1-ethyl-3-methylimidazolium bis(trifluoromethylsulfonyl)imide ([EMIM][TFSI]) ionic liquid (IL), poly(ethylene glycol) diacrylate (PEGDA) monomer, and a 1-hydroxycyclohexyl phenyl ketone photoinitiator as a chemoreceptive material as well as gate electrolyte layer. PEDOT:PSS has been widely described as an ion-permeable p-type conducting polymer in OECTs due to its tunable conductivity, ease of deposition, and use in flexible and stretchable devices[37–39]. Because ILs, including [EMIM][TFSI], possess high thermal and chemical stability, non-volatility, and high carrier-inducing and gas-solvating abilities, they can be used as gas-sensing layers in electrochemical or potentiometric gas sensors[40–42]. An ACNS based on an OECT can emulate neurotransmitter release (in the form of ions) from the presynaptic membrane (IG electrolyte), resulting in excitation or inhibition of the postsynaptic membrane (PEDOT:PSS channel) in response to an electrical or chemical external stimulus. Due to ion permeation of the channel layer, high volumetric capacitance allows OECTs to operate at relatively low voltages suitable for emulating biological synaptic functions. Nitrogen dioxide ($NO_2$) gas was used as a potentiating chemical stimulus, and electrical gate pulsing was used as an inhibiting stimulus to drive the ions between the presynaptic and postsynaptic membrane. Upon exposure to $NO_2$ gas, the ACNS showed a stable response toward different concentrations of $NO_2$ and extended retention of the changes in an excitatory postsynaptic current (PSC), i.e., synaptic weight (SW), which demonstrates long-term memorization. The ACNS is gated by the potential generated by the interaction between $NO_2$ molecules

with cations within the chemoresponsive ionogel. The potentiation effect of chemical gas pulsing can be impeded by applying inhibitory electrical pulses, changing the overall PSC of the postsynaptic membrane. This report provides a description of an artificial chemosensory synapse for artificial olfactory systems in which the IG layer responds to chemical and electrical stimuli and, in turn, contributes to the modulation of the PSC, exhibiting both potentiation and inhibition. The ACNS demonstrated an ability to mimic biological synapses between ORNs and PNs as well as interneurons, and to retain ions in the channel for an extended period, resulting in long-term memorization of the stimuli.

## Results

Biological synapses in the olfactory system form between the axons of olfactory receptor neurons (ORNs) and the dendrites of two projection neurons (PNs), i.e., mitral and tufted cells, in a spherical structure called the glomerulus, which is located in the olfactory bulb. The ORNs articulated with the same odor-binding proteins release excitatory neurotransmitters (glutamate) in the same synaptic zone in the glomerulus[31,43]. Two inhibitory interneurons of periglomerular cells (PGCs) and granule cells (GCs) form synapses inside the glomerulus with PNs (Fig. 1a). The PGCs release the inhibitory neurotransmitter gamma-aminobutyric acid (GABA) in the glomerulus, forming inhibitory synapses with PNs[32,34]. By modulating the presynaptic signals, these synapses can produce excitatory or inhibitory postsynaptic signals and convey complex details to the postsynaptic receptors activated by respective neurotransmitters, resulting in overall learning and forgetting or modulation of signals in a single device. By simply emulating the functions of these neurons, an artificial chemoreceptive system can be built and utilized as a component of bio-inspired sensor signal processors that exhibit sensing, learning, forgetting, and modulating signals for intrinsically intelligent olfactory systems[44].

Figure 1b depicts a schematic of the proposed ACNS, based on an OECT that can be gated by both chemical and electrical stimuli and function with excitatory and inhibitory action potentials, respectively, for the ACNS. Details of the OECT fabrication process for the ACNS are described in the Materials and Methods section. OECTs have been widely studied for possible use in bio-interfaces, chemical and biological sensing, and synaptic devices[18,19,21,45]. Stimuli (either electrical or chemical) in an OECT can cause an IG to inject or extract cations from the p-type PEDOT:PSS channel, inducing de-doping or doping in the channel, respectively[46]. Subsequent modulations in PSC (ΔPSC), which translate to potentiation and inhibition, can be realized by applying chemical and electrical stimuli, respectively (Fig. 1c). Due to the nature of the gating effect in the ionogel under chemical stimulation, potentiation of the OECT synapse can be obtained by chemical pulsing without accompanying electrical gate pulsing. In this study, the pulse duration time ($T_{on}$) and the number of pulses ($P_n$) of the applied stimuli were varied. This phenomenon resembles neurotransmitter transport across a biological synapse when an action potential is transported through the cleft of the synapse.

Figure 1d provides a comparative breakdown of the components of the biological chemosensory synapse in the olfactory system and ACNS for artificial olfactory systems. The channel forms the synaptic cleft of the artificial chemical synapse, similar to the synaptic cleft between the presynaptic ORNs and postsynaptic PNs. The IG corresponds to the presynaptic membrane in ORNs, receiving the stimulus, and the drain electrode in the OECT acts as a postsynaptic membrane in the mitral/tufted cells (PNs) and collects the PSC (i.e., drain current, $I_d$) for further signal processing. When a chemical stimulus in the form of an $NO_2$ gas pulse is applied to the IG, anions, behaving as an excitatory neurotransmitter (glutamate) diffuse into the channel layer, increase the PSC of the device (Fig. 1c). The gate electrode forms an inhibitory synapse with the channel layer, acting as an inhibitory presynaptic membrane. The electrical stimulus applied to the gate electrode controls the flow of

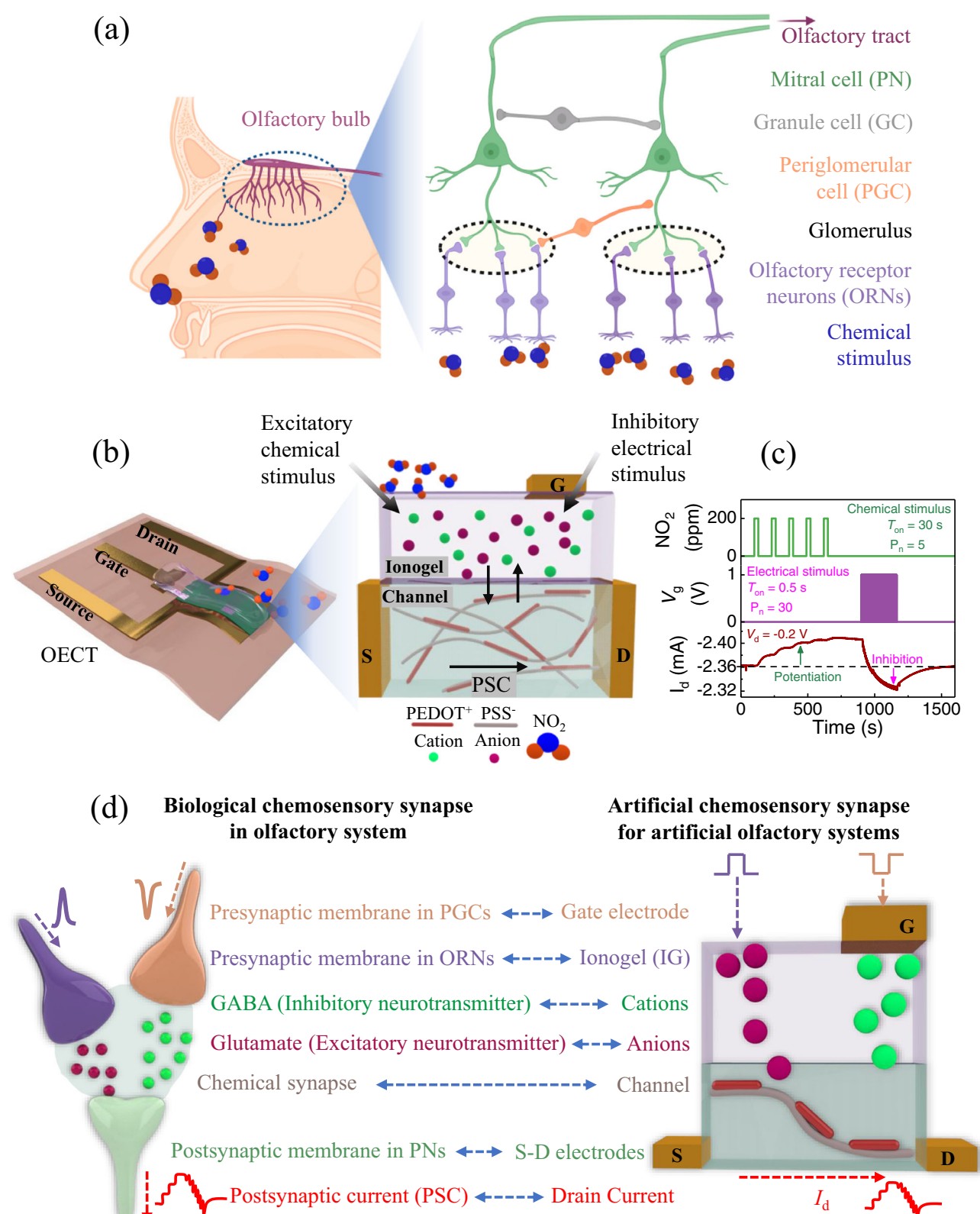

ions to the channel, playing a role in controlling an inhibitory neurotransmitter (GABA) and impeding the ΔPSC induced by excitatory chemical stimuli.

Chemical synapses release neurotransmitters in the synaptic cleft by opening ion channels when an action potential applied to a presynaptic membrane transmits signals from the presynaptic to the postsynaptic membrane (Fig. 2a)[47]. A similar approach can be used to

describe an electrically regulated artificial chemical synapse (Fig. 2b). To characterize the device under an inhibitory electrical stimulus, we assessed the comparative roles of IL and IG electrolytes in the OECT in sensing and their synaptic properties. We fabricated and tested different IG compositions (Supplementary Fig. S1a), and established from the transfer curves that an IG with 90 wt.% IL achieved a comparatively lower threshold voltage ($V_{th}$) and higher on-off current ratio ($I_{on}/I_{off}$)

**Fig. 1 | A biological olfactory system and the concept of an artificial chemosensory neuronal synapse (ACNS). a** Synaptic connections between the olfactory receptor neurons (ORNs), mitral cells (PNs), and interneurons form inside the glomerulus in the olfactory bulb. The synapses can be excitatory or inhibitory. The subsequent postsynaptic current (PSC) is conveyed to the olfactory tract for further processing. **b** An ACNS based on an organic electrochemical transistor (OECT) with an ionogel as the sensing layer as well as the gate electrolyte. Ions in the ionogel are injected or extracted from the organic semiconductor channel,

resulting in de-doping or doping phenomena. **c** Chemical ($NO_2$) and electrical ($V_g$) stimuli are applied as presynaptic excitatory and inhibitory action potentials, respectively, which can result in ion exchange between the channel and electrolyte, modulating the $I_d$ i.e., the postsynaptic current (PSC). The graph shows the PSC variation with different stimuli, where $T_{on}$ is the pulse duration time and $P_n$ is the number of pulses of the applied chemical and electrical stimuli. **d** A comparative breakdown of biological and artificial chemosensory neuronal synapse components.

---

(Supplementary Fig. S1b). The IGs with lower IL content (40 and 60 wt.% IL) resulted in a harder solid structure whereas higher IL content in the IG (90 wt.% IL) can help form a soft gel-like structure of IG (Supplementary Fig. S1c). The fundamental transfer and output characteristics of IL- and IG (with 90 wt.% IL)-gated OECTs (Supplementary Fig. S2a–d) demonstrated device operation in depletion mode. The transfer curves of six IG (90 wt.% IL)-gated devices fabricated on the same substrate show good reproducibility in their transfer curves (Supplementary Fig. S2e) and also their estimated $V_{th}$ and $I_{on}/I_{off}$ values (Supplementary Fig. S2f, g). When the gate pulsing was applied in the form of positive gate voltage ($V_g$), the cations in the electrolyte were pushed into the PEDOT:PSS channel, lowering the channel conductance from its original doped state ($PEDOT^+$) (Fig. 2b, schematic I). This depleted the number of holes in the channel, resulting in de-doping of the original $PEDOT^+$ in the polymer chain and creating neutral $PEDOT^0$ (Fig. 2b, schematic II). When $V_g$ was removed, the cations slowly diffused back to the electrolyte, returning the neutral $PEDOT^0$ to its original doped state ($PEDOT^+$) (Fig. 2b, schematic III). This shows that the application of a positive $V_g$ will drive the device into a low-conductance state from an initially high-conductance state.

The conductance state of the device after gate biasing and the rate of the recovery process of the cations in the channel are the essential parameters that control and directly influence the synaptic plasticity of the ACNS. Synaptic plasticity implies activity-dependent changes in synaptic transmission between presynaptic and postsynaptic membranes, whereas SW is the strength of the connection between the two membranes due to synaptic activity. To understand the synaptic properties of our device under gate biasing, we investigated the synaptic functions, such as paired-pulse depression (PPD), spike amplitude-dependent plasticity (SADP), spike duration-dependent plasticity (SDDP), and spike number-dependent plasticity (SNDP), which are the basic properties of artificial synaptic systems. To measure the changes in source-drain current ($I_d$), which is the equivalent of PSC, the source-drain voltage ($V_d$) at the postsynaptic membrane was fixed at −0.2 V throughout the investigation, which is low enough not to obstruct the ion dynamics. Figure 2c shows the PSC variation of the ACNS when two consecutive $V_g$ pulses of 1 V (pulse width of 0.5 s) were applied on the IG (90 wt.% IL)-gated OECT, emulating the PPD. The SW calculated by $|(PSC_{SW} - PSC_{in})/PSC_{in}|$ after 30 s of termination of the applied gate pulses was 6.8%, where $PSC_{in}$ and $PSC_{SW}$ are the PSC values before and after applying the gate pulses where the SW is calculated, respectively.

After applying a stimulation, PSC decays to its initial state rapidly or slowly, depending on the retention behavior of the ions in the channel, which is usually referred to as short- or long-term plasticity (STP or LTP, respectively). The number of neurotransmitters released into the synaptic cleft can vary depending on the presynaptic membrane potential. Application of repetitive, prolonged, and large action potentials can convert STP to LTP in chemical synapses. To demonstrate similar properties in an ACNS, we investigated the SADP function of both IL- and IG-gated devices by applying a $V_g$ pulse with an amplitude of 1 V (pulse width of 0.5 s) (Fig. 2d). It was evident that the ΔPSC in the IG (90 wt.% IL)-gated device was larger than in the IL-gated device. To further investigate the synaptic behavior, we characterized SDDP characteristics of the devices (Fig. 2e). A single $V_g$ pulse of 1 V

with a pulse duration of 5 s was applied to both IL- and IG (90 wt.% IL)-gated devices to estimate ΔPSC. The SNDP characteristics of the devices were also determined by applying repetitive $V_g$ pulses with a fixed amplitude and pulse width (1 V and 0.5 s, respectively) (Fig. 2f). The SW values of the devices were determined for SADP, SDDP, and SNDP functions. The high retention of the ΔPSC state after applying the electrical stimuli affected the SW values of the devices, which was confirmed by increasing the $V_g$ pulse amplitude (0.5, 0.8, 1, 1.2, and 1.5 V), while keeping the pulse width constant at 0.5 s to estimate the dependence of PSC on pulse amplitude, i.e., SADP (Fig. 2g). The detailed time-dependent PSC of IL- and IG (90 wt.% IL)-gated devices showing SADP can be found in Supplementary Figs. S3a and S4a, respectively. The SW recorded after 30 s of termination of $V_g$ pulsing gradually increased from 0.5% to 5.4% for IL-gated devices and 0.7–10.9% for IG-gated devices. The overall increase in the SW value after increasing the amplitude of the $V_g$ pulses indicates that a higher-magnitude $V_g$ can drive more cations into the channel, resulting in longer retention times and higher SW values, and converting STP to LTP. These results indicate that IG-gated devices can achieve a higher SW compared with IL-gated devices. Similarly, the PSC recorded by increasing the pulse width from 0.1 to 5 s to emulate SDDP function (Supplementary Figs. S3b and S4b for IL- and IG-gated devices, respectively) resulted in an overall increase in the SW for the pulse duration in both IL- and IG-gated devices (Fig. 2h). Due to the repetitive stimulation of the device (SNDP), the PSC increases with the number of electrical stimuli ($V_g$ pulses with 1 V of amplitude and 0.5 s of width), as shown in Supplementary Figs. S3c and S4c for both the IL- and IG (90 wt.% IL)-gated devices, respectively. The estimated SW values increased up to 70% in the IG-gated device and 20% in the IL-gated device (Fig. 2i), respectively, demonstrating LTP at a greater number of applied $V_g$ pulses when the number of electrical stimuli increased from 1 to 50.

The higher retention time and the slow recovery of the PSC to its initial state also contributed toward higher SW values. To demonstrate the retention of PSC state after electrical stimulation, the decay time constant ($\tau$) of the device was analyzed by employing the exponential decay function. Supplementary Fig. S5 shows the analyzed decay time constants for both IL- and IG (90 wt.% IL)-gated devices at different magnitudes of the electrical stimulus. It is established from the decay constant estimation that IG (90 wt.% IL)-gated device shows longer retention times compared to IL-gated device (up to $19 \times 10^3$ s at stimulation amplitude, $V_g = 1.5$ V) and slow decay of PSC to its initial state, generating higher SW values. Increased SW and τ values along with shift from STP to LTP was observed, by increasing the magnitude of electrical stimuli.

The results provided in Fig. 2 indicate that, with repetitive, prolonged, and higher magnitude of stimulation, we could observe LTP in ACNS, akin to chemical synapses in the SNS. The IG-gated device achieved a higher SW in all cases. This can be attributed to the slow redistribution of ions in networked IL channels inside the polymer matrix of an IG after the electric field across it is removed. Optical microscopy confirmed a uniformly deposited IG (~150 μm thick) (Supplementary Fig. S6). The dropped IL exhibited a stability problem, presumably due to hydrophobicity, instability due to the flow of IL on the device surface, and difficulty establishing a uniformly thin layer.

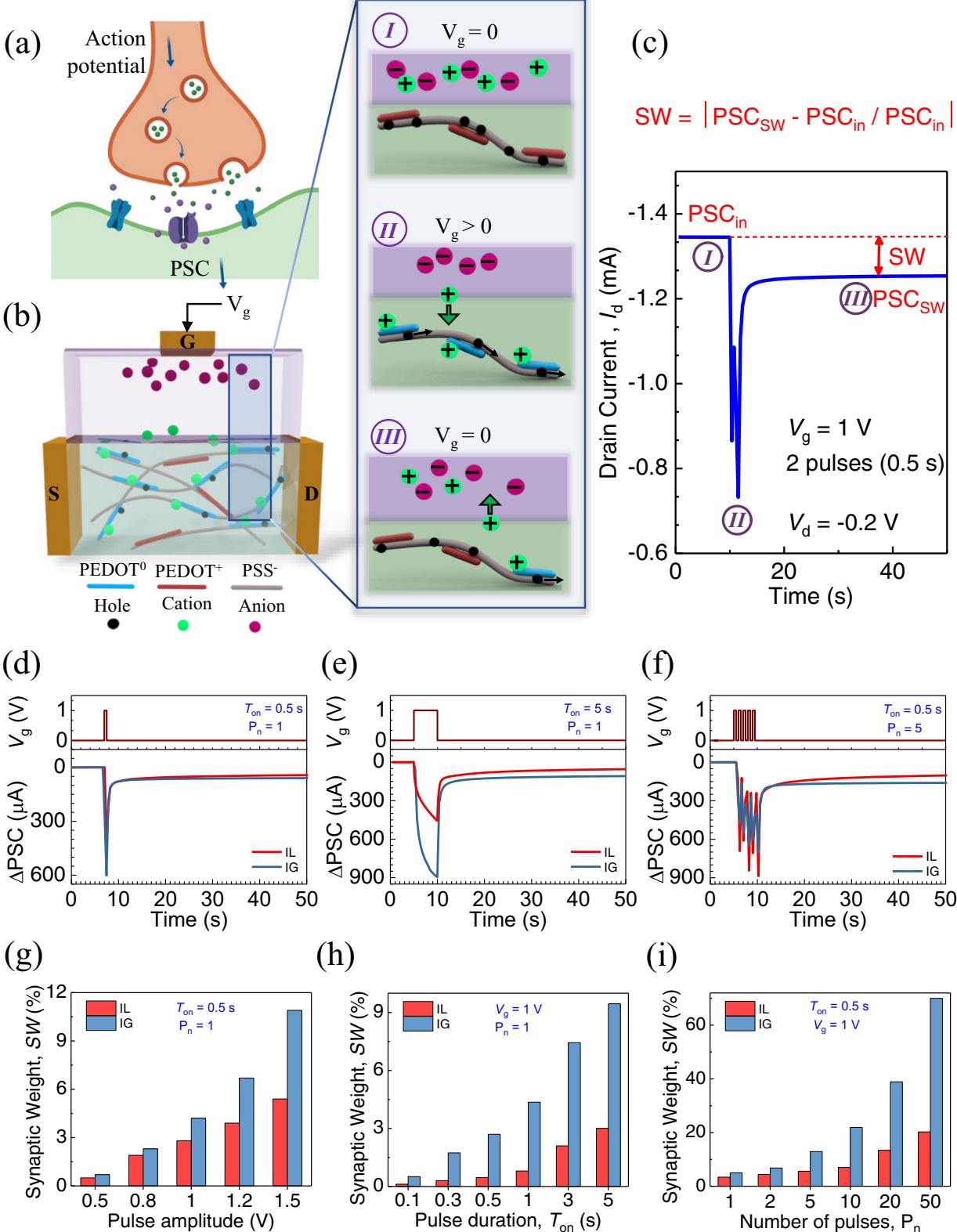

The longer retention time of the PSC and higher SW in the IG-gated devices is presumably the result of the blocking effect created by the highly networked porous structure in the polymer matrix of IG, which is believed to slow the redistribution of ions in an IG upon removal of the electric field. These results suggest that the liquid nature of ILs limits their use in solid-state devices. In contrast, IGs possess a solid-state nature and ionic conductivity, which is beneficial for stable device operations. Because our device is fabricated on a flexible polyimide substrate, we also tested the electrical properties of IL- and IG-gated devices under mechanical bending, in which the measured transfer curves showed evidence of instability in IL-gated devices upon bending compared with IG-gated devices (Supplementary Fig. S7a, b). The time-dependent $\Delta PSC$ of the IL-gated device upon bending was not significant under no gas exposure, but resulted in a larger abrupt

**Fig. 2 | Electrical regulation of the ACNS. a** Schematic of a biological chemical synapse. When an action potential is generated, neurotransmitters are released from the presynaptic membrane to the postsynaptic membrane. **b** Schematic diagram of the artificial chemical synapse. The inset shows the operating mechanism of the artificial synapse, (I) before an action potential ($V_g$ pulse) is applied, and (II) when an action potential ($V_g > 0$) is applied to the presynaptic gate electrode, the ions diffuse from the electrolyte to the channel which results in a change in ΔPSC, which can be observed between source and drain electrodes. (III) Ions slowly diffuse to their original distribution once the action potential is removed. **c** PSC variation and an explanation of SW in an artificial chemical synapse when two consecutive $V_g$ pulses are applied, corresponding to the operation mechanism described in (**b**). **d–f** Time-dependent measurement of ΔPSC for SADP, SDDP and SNDP properties. Changes in SW were measured for the devices with ionogel (IG with 90 wt% IL) and ionic liquid (IL) electrolytes by varying the (**g**) amplitude, (**h**) duration, and (**i**) number of electrical stimuli (i.e., gate pulses). The source-drain voltage ($V_d$) i.e., the reading voltage, is maintained at −0.2 V for all cases.

change in the ΔPSC upon gas exposure, in comparison with the ΔPSC of the IG-gated device (Supplementary Fig. S7c) presumably due to the instability of IL in the device under a bended condition (Supplementary Fig. S7d).

In the SNS, multiple chemical synapses formed between the ORNs and PNs are controlled by varying the electrochemical excitation (known as an action potential), which plays a large part in perception. When a chemical stimulus (in the form of gas molecules) is applied to the ORNs, the ORNs sense and sequentially transport a corresponding action potential through the neuron, which releases neurotransmitters via exocytosis into the synaptic cleft, and the resulting changes in PSC are transported by PNs to higher olfactory centers for further processing (Fig. 3a). The schematic in Fig. 3b illustrates the ACNS concept, in which the device mimics a synaptic connection between the ORN and PN in an olfactory system. To emulate the concept, our ACNS establishes a chemoreceptive function for chemical stimulation using an IG that can respond to the chemical properties of a gas. We checked the device responses under four different analytes ($NO_2$, $NH_3$, $SO_2$, and acetic acid) as shown in Supplementary Fig. S8 and continued the further experiments with $NO_2$. Upon exposure to $NO_2$ gas, the gas molecules interact with cations in the IG, which drives ions into the channel in the manner of a neurotransmitter, resulting in the modulation of PSC.

Transfer curves were measured to determine the basic characteristis of the device upon gas exposure. They showed an increase in the on-state current and a large positive shift in the $V_{th}$ of both IG (90 wt.% IL)- ($\Delta V_{th} = 1.6$ V) and IL-gated devices ($\Delta V_{th} = 0.75$ V) when exposed to 400 ppm of $NO_2$ (Fig. 3c). Changes in $V_{th}$ of the IG-gated device at varying concentrations of $NO_2$ are shown in Supplementary Fig. S9a. The $\Delta V_{th}$ was also observed under chemical exposure without electrical gate biasing. A positive shift in $V_{th}$ after gas exposure may be attributable to the increasing number of anions in the channel. The presence of $NO_2$ can consequently induce a negative gating effect at the IG, resulting in an increase of PSC. The transfer curves show a larger $V_{th}$ shift in the IG-gated device, which indicates that more anions penetrated the channel layer compared with the IL-gated device when both devices were exposed to the same concentration of $NO_2$ for the same duration of time. This is presumably due to the faster movement of ions in the thin layer of the IG when an electric field is present across it. The spin-coated and cured IG forms a thin layer on the device compared with the same amount of IL (Supplementary Fig. S6). We also compared the responses of an IG (90 wt.% IL)-gated OECT and an OECT without IG (Supplementary Fig. S9b). Compared to the IG (90 wt.% IL)-gated OECT, the device without IG showed a small, recoverable response to 400 ppm $NO_2$. The results indicate that the IG (90 wt.% IL)-gated OECT structure can be utilized to get a high response with high retention of the induced changes in PSC via the chemical-gating effect.

To demonstrate the emulation of the chemoreceptive and synaptic functions in our ACNS and understand the operating mechanism, time-dependent ΔPSC data under chemical gating were measured. The dynamic ΔPSC data of the IL- and IG (90 wt.% IL)-gated devices was measured at $V_d = -0.2$ V with different concentrations, durations, and successive periods of exposure to the chemical ($NO_2$ gas) pulses. The results showed an increase in ΔPSC when the concentration of the $NO_2$ pulse (with a duration of 30 s) was increased from 100 to 600 ppm (Fig. 3d). The higher the concentration, the stronger the response. Similarly, the dependence of ΔPSC on the chemical pulse duration (Fig. 3e) and the pulse number (Fig. 3f) indicates increases in the response, with increasing the duration and number, respectively. This was confirmed by the estimated SW after 300 s for the dependence of ΔPSC on the pulse concentration, duration and number. The SW value increased from 0.04 to 3% in the case of the IL-gated device and 1.2 to 16% for the IG-gated device when the $NO_2$ concentration increased from 200 to 600 ppm (Fig. 3g). Similarly, a gradual increase in SW was observed upon increases of the pulse duration (Fig. 3h) and number (Fig. 3i).

In all cases, the IG-gated device achieved a higher ΔPSC compared with the IL-gated device, which can be attributed to the more effective diffusion of ions from the thin IG to the channel due to $NO_2$ absorption. Chemical pulses with a longer exposure time (width of pulse), when applied continuously (with a short resting time between the pulses), may saturate the $I_d$ of the device, indicating a limited absorption capacity of the IG under long-term gas exposure (Supplementary Fig. S10a). Similarly, the PSC variation of the device indicates a saturated state when the concentration of $NO_2$ pulse increases in a step-like manner with time (Supplementary Fig. S10b). Concentration-dependent changes in PSC (i.e., ΔPSC) of ACNS for lower concentration $NO_2$ (10, 30, 50, and 100 ppm) with a fixed pulse duration of 30 s (Supplementary Fig. S10c) were obtained and the limit of detection of the device was calculated as low as 2.66 ppm from the data (Supplementary Fig. S10d)[48]. Also, to study the dependency of ΔPSC on the pulse duration, the ΔPSC was estimated for 10 and 30 ppm of $NO_2$ with varying pulse duration (20, 30, and 60 s), as shown in Supplementary Fig. S10e, S10f, respectively. Both studies showed a gradual increase in the ΔPSC with increasing the concentration and duration of $NO_2$ pulses for a fixed duration and concentration, respectively.

In a conventional gas sensor, the device recovers to its original state once it is no longer exposed to the gas. Time-dependent PSC measurements with different concentrations, durations, and the number of chemical pulses (Fig. 3d–f) indicate that the increase in the PSC is preserved after termination of gas exposure and does not recover to its original state for hundreds of seconds. We observed long-term retention of the ΔPSC, resulting in long-term memorization of the input stimuli, even after the termination of gas pulsing. This unique behavior has rarely been reported and may be attributable to the solvation of $NO_2$ in [EMIM][TFSI][49,50] (Fig. 3b). The Lewis structure of $NO_2$ is consistent with the presence of an unpaired electron, which makes it susceptible to reactions. In a stable state, $NO_2$ readily forms a dimer ($N_2O_4$) in the IL/IG, lowering the energy of the entire system[51,52]. The mechanism of $NO_2$ absorption in [EMIM][TFSI] has been reported by several groups[42,49], indicating a strong tendency for $NO_2$ to be dimerized in the stable state. Dimerized $N_2O_4$ tends to form π–π interactions with the imidazole cation, [EMIM]$^+$ in the IL/IG (Fig. 3b). Because the $NO_2$ molecules interact with the cations in the IL/IG, the anions presumably diffuse into the channel layer due to the negative gating effect created in the IG and interact readily with PEDOT$^+$ for stable charge redistribution, making room for more $NO_2$ molecules in the electrolyte. This may result in an overall increase in PSC due to the doping effect in the channel. The phenomenon of penetration of

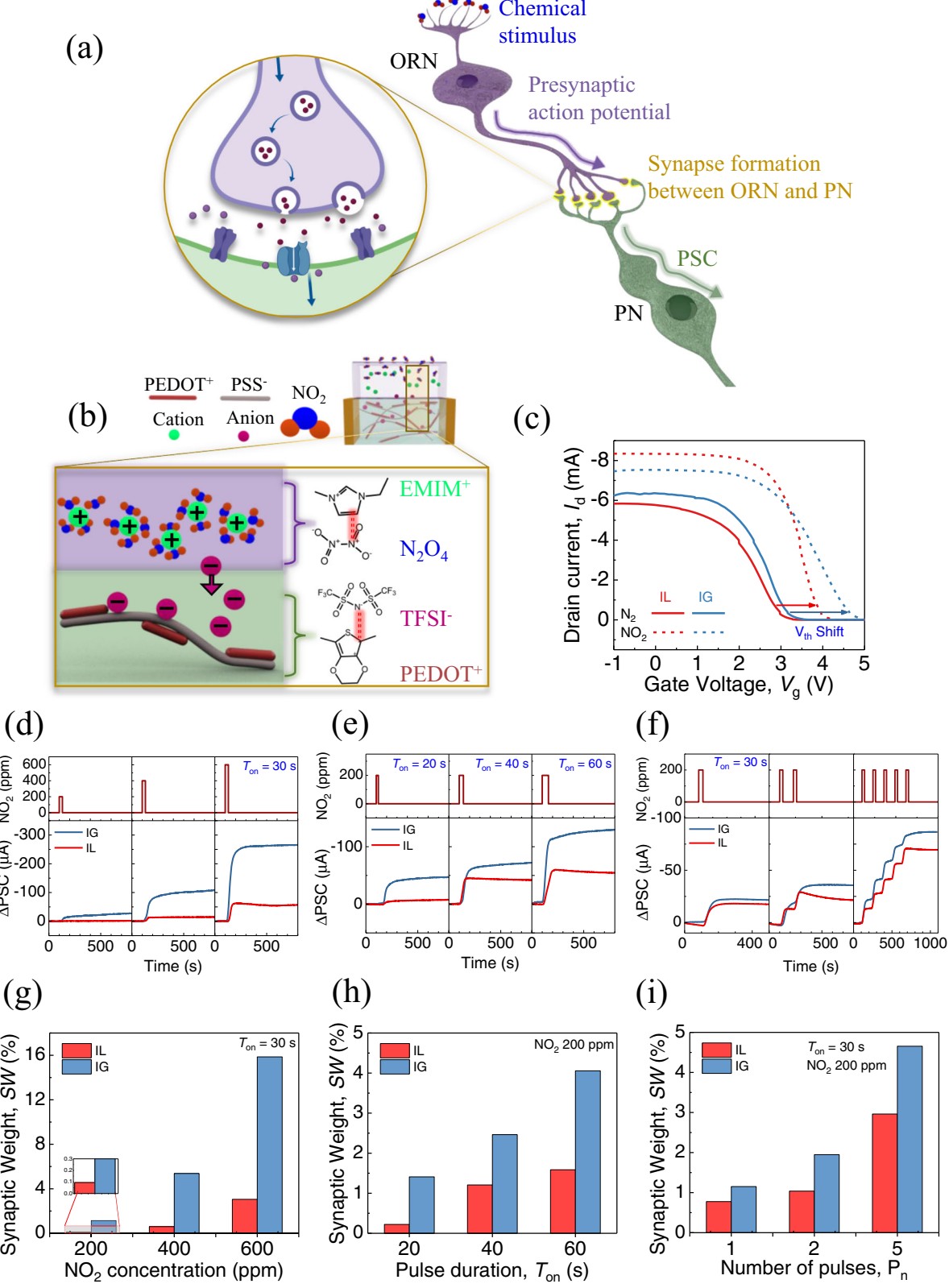

anions into the channel and an increase in the ΔPSC can be also observed by applying a negative electrical pulsing at the gate (Supplementary Fig. S11). The results indicate that the anions indeed diffuse into the channel upon chemical stimulation of the device. Unlike the electrical stimulation, the anions are retained in the channel for an extended period upon chemical stimulation, indicating a strong interaction between the solvated complex of cations and $NO_2$ in the IL/IG,

hindering the diffusion of the anions in the channel to their initial positions. The potentiation mechanism by chemical stimulation can therefore be explained by the chemically driven gating effect, which is induced by the potential in the IG generated by interactions among $NO_2$ molecules and cations. Because $NO_2$ molecules form stable bonds with cations in the IL/IG, it is difficult to desorb the gas from the IL/IG without external aid, resulting in long-term retention of the charge carriers in the

**Fig. 3 | Chemical regulation of the ACNS. a** A biological chemical synapse with chemical stimulus. A chemical stimulus (in the form of gas molecules) is sensed by the ORN, and a corresponding action potential is then transported through the neuron, releasing neurotransmitters via exocytosis into the synaptic cleft. The resulting changes in ΔPSC are transported by PN to higher olfactory centers for further processing. **b** An artificial chemical synapse with chemical stimulus. $NO_2$ dimerizes to form $N_2O_4$ in the stable state, facilitating π−π interactions between the $[EMIM]^+$ cations and $N_2O_4$. Cations in the electrolyte are solvated by gas molecules, resulting in diffusion of $[TFSI]^-$ anions into the channel to interact with $PEDOT^+$, increasing the ΔPSC due to gating inducement of the doping effect. **c** Transfer

curves show a positive shift in threshold voltage ($V_{th}$) upon exposure to $NO_2$ gas (400 ppm). The IG (90 wt.% IL)-gated OECT shows a larger positive shift and greater gas absorption. **d** Time-dependent measurement of the ΔPSC with different concentrations (200, 400, and 600 ppm) of $NO_2$ with $T_{on}$ = 30 s. The corresponding SW is presented in (**g**). **e** Time-dependent measurement of ΔPSC with different duration times, i.e., $T_{on}$ of $NO_2$ (200 ppm) exposure. The corresponding SW is quantified in (**h**) for both IL- and IG-gated devices. **f** Time-dependent measurement of ΔPSC with different numbers of $NO_2$ (200 ppm) pulses i.e., $P_n$ (1, 2, and 5 pulses) indicates a gradual increase in SW, which is depicted in (**i**).

channel, which implies long-term memory. Inhibition of PSC can be achieved by applying an external electrical field, which can mitigate the chemically driven gating effect, resulting in the extraction of anions in the channel and injection of cations freed from the solvated complex of $NO_2$ gas molecules and cations.

The schematic representation of our ACNS mimicking of the functions of a chemical synapse with excitatory or inhibitory presynaptic membranes in the biological olfactory system (Fig. 4a), is shown in Fig. 4b, describing the working principle when chemically exciting and electrically inhibiting stimuli are applied. The excitatory chemical stimulus drives the anions from the electrolyte into the channel layer, resulting in an increase in the PSC due to the doping effect in the p-type organic semiconductor channel. An electrical stimulus followed by a chemical stimulus will force these anions to diffuse out of the channel back into the electrolyte layer and instead drive the cations into the channel layer, resulting in a decrease of PSC due to de-doping in the channel. This diffusion phenomenon of cations and anions into and out of the channel layer under different inducements mimics the behavior of neurotransmitters in olfactory inhibitory and excitatory chemical synapses.

To mimic the functions of excitatory and inhibitory synaptic events in the glomerulus, we first tested the response of the ACNS to excitatory chemical stimuli followed by inhibitory electrical stimuli (Fig. 4c). The IG (90 wt.% IL)-gated device was selected for further measurements (for comparison of IL- and IG-gated devices, see Supplementary Fig. S12). The ΔPSC is shown with excitatory chemical pulses resulting from varying the $NO_2$ concentration (100, 200, 400, and 600 ppm, with a chemical pulse duration of 30 s), followed by inhibitory electrical pulses (a $V_g$ amplitude of 1.3 V with an electrical pulse duration of 0.5 s). As the concentration of the excitatory chemical stimulus, i.e., $NO_2$ concentration, increased, the overall ΔPSC value was enhanced. Because the device exhibited a long-term memory effect when exposed to the gas and did not recover to its initial state for an extended period, this property can be used to depict several memory states of the device after each successive chemical pulsing. The ACNS did not show significant degradation in the memory state after the termination of gas pulses and maintained the PSC level even after a few hundred seconds. The ΔPSC level changed when followed by 200 continuous inhibitory electrical pulses of positive $V_g$ (with a duration time of 0.5 s). The inhibitory electrical pulsing of positive $V_g$ decreased the ΔPSC level, and at lower $NO_2$ concentrations of 200 and 400 ppm was able to bring the ΔPSC level back to the initial state. For a higher concentration of excitatory $NO_2$ pulses, a higher voltage of inhibitory electrical pulse amplitude ($V_g$) pulses would be required for the ΔPSC to recover to its initial state, which means a higher electrical potential would be required to deplete the cations from the channel layer and turn off the device (consequently extracting the anions). Further demonstration of the time-dependent ΔPSC of the IG-gated device, with a fixed excitatory chemical stimulus followed by inhibitory electrical stimuli of different pulse amplitude ($V_g$), pulse duration ($T_{on}$), and pulse number ($P_n$), is shown in Supplementary Fig. S13.

Measurement of the activity-dependent changes in synaptic plasticity in the synapses of the SNS can help predict postsynaptic signal enhancement or decrement under dynamic stimuli. The order of and the interval between the excitatory and inhibitory action potentials in the synapses can influence the overall postsynaptic signals. To predict the changes in the SW due to the time interval between two different stimulations, the effect of excitatory chemical pulsing followed by inhibitory electrical pulsing on the change in the ΔPSC, with different time intervals between the two events, is shown in Fig. 4d. In contrast, Fig. 4e displays the reverse situation, i.e., inhibitory electrical pulsing followed by excitatory chemical pulsing. Anions and cations diffuse into the channel during the excitatory and inhibitory events, respectively, mimicking excitatory and inhibitory neurotransmitter release in the synaptic cleft. A short interval between the two events in the biological synapses indicates the presence of neurotransmitters from the first synaptic event in the synaptic cleft when the neurotransmitters from the second event are released, affecting overall synaptic plasticity. Similarly, in the case of ACNS, anions, and cations can be present concurrently in the channel if the excitatory and inhibitory synaptic events occur at sufficiently short time intervals. As a parameter describing the different time intervals between the two events, we defined the difference in stimulation time between the excitatory (chemical pulsing) and inhibitory (electrical pulsing) events as Δt (Δt = $t_2$ − $t_1$, where $t_1$ is the ending time of the first stimulation, and $t_2$ is the starting time of the second stimulation). An increase in Δt between the excitatory and inhibitory events, as demonstrated in Fig. 4d, e (from bottom to top), caused changes in ΔPSC and then SW. The overall normalized change in SW (calculated at 4τ interval after the two stimulations in all cases) vs. Δt (the difference in the stimulation time between the excitatory and inhibitory events) was calculated and is presented in Fig. 4f (for quantitative values, see the data in Supplementary Fig. S14). When an excitatory chemical stimulus was applied ahead of the inhibitory electrical stimuli, a gradual decrease in SW was observed with increasing Δt, and the opposite trend was seen when inhibitory electrical stimuli were applied ahead of the excitatory stimulus. This behavior can be attributed to the fact that, when the inhibitory pulses are applied immediately after an excitatory pulse, i.e., Δt = 5 s (red in Fig. 4d), the ions do not have enough time to achieve a stable equilibrium state in the device. The inhibitory electrical pulses would not be able to expel a large number of anions from the channel layer as the anions from the IG are still inside the channel due to the gradual interaction of gas molecules and cations. In comparison, if Δt is large (160 s) (green in Fig. 4d), the ion distribution can converge on a stable equilibrium state after the gas absorption, and ions can be extracted from the channel more efficiently by applying inhibiting electrical pulses. At very high Δt (260 s), the influence of first stimulation on the second stimulation would be almost negligible. However, when inhibitory electrical stimuli were applied ahead of an excitatory chemical stimulation (Fig. 4e), a gradual increase in the SW with increasing Δt were observed. When Δt is small (red in Fig. 4e), the decrease in ΔPSC due to inhibitory electrical pulsing will be compensated for immediately by the excitatory chemical pulsing, making the

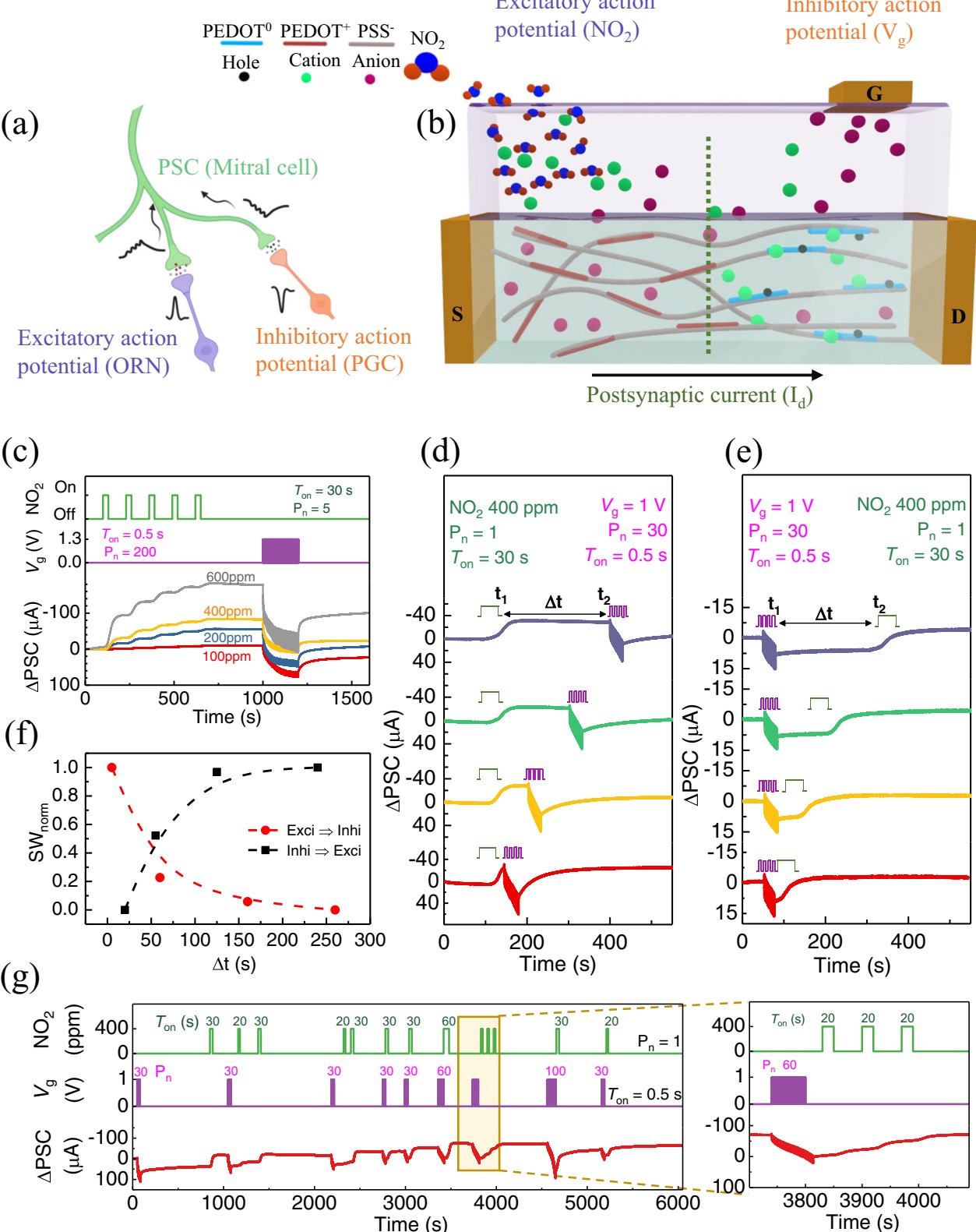

**Fig. 4 | Excitatory and inhibitory functions of the ACNS. a** Chemical synapses inside the glomerulus acquire different action potentials from ORN and PGC and transfer the modulated postsynaptic signals to the olfactory bulb. **b** Schematic of the ACNS, mimicking the biological olfactory system and showing the excitatory and inhibitory functions. The chemical stimulus contributes to excitatory PSC by diffusion of anions, whereas the electrical stimulus results in PSC inhibition by driving cations into the channel. The net modulated PSC is collected at the source-drain electrodes. **c** The ACNS with pulsed chemical stimuli at different concentrations followed by an equal number of electrical pulses, showing potentiation first and then inhibition functions. The changes in PSC with (**d**) excitatory followed by inhibitory and (**e**) inhibitory stimuli followed by an excitatory stimulus with different time intervals (Δt) between the two types of stimuli. The consequent normalized SW is calculated and presented in (**f**). **g** Under random excitatory and inhibitory stimuli, the ACNS shows gradual modulation in the ΔPSC according to the time and number of applied pulses.

net change in ΔPSC and SW small. At larger Δt (purple in Fig. 4e), cations that penetrate the channel due to the electrical stimuli will have enough time to diffuse back to the IG, and the excitatory chemical stimulus will further increase the conductance state, increasing the overall ΔPSC and SW. This can help predict changes in the SW due to the time interval between the different stimulation events.

The ACNS was tested over an extended period of 6000 s to determine the operational stability and responses of the device under random stimulations (Fig. 4g). Random excitatory and inhibitory pulses of varying pulse durations and numbers were applied, respectively, with different intervals between the successive stimuli. The device indicated a change in the ΔPSC with different excitatory chemical and inhibitory electrical stimuli of 400 ppm $NO_2$ and $V_g$ of 1 V, respectively. Continuous measurement over 6000 s revealed the ability of the device to repeatedly carry out the functions of long-term memorization through excitatory chemical pulses and depression through inhibitory electrical pulses. We successfully emulated the excitation and inhibition functions of chemical synapses in an artificial olfactory system, leading to sensing, memorization, forgetting, and overall modulation of the synaptic signals.

To demonstrate the capability of the ACNS for an application to the data-efficient bio-inspired perception of input stimuli, the SW values from the time-dependent response (ΔPSC) of IG (90 wt.% IL)-gated OECT under chemical stimulation ($NO_2$ with $T_{on}$ = 30 s) with varying the concentration from 100 to 400 ppm (Supplementary Fig. S15a) were used for training and prediction of the concentration using a machine learning (ML) algorithm. Here, for the purpose, we measured the performance of the ACNS eight times at each concentration (total of 32 samples) for the classification of different concentrations of $NO_2$ (ranging from 100 to 400 ppm) utilizing the support vector machine (SVM) in Classification Learner application in the ML toolbox available in the MATLAB software. Instead of utilizing a full-time-dependent variation of ΔPSC of the device for ML, as an example, we considered the SW values at 3 instances (at 90, 210, and 330 s at equally spaced time interval of ~120 s) after the chemical stimulation ($NO_2$ with $T_{on}$ = 30 s) for generating the dataset for ML. The results were shown to have test accuracy of 81.2% (Supplementary Fig. S15b). From the results, it can be concluded that four different concentrations of $NO_2$ can be classified successfully after training, even utilizing significantly fewer training and test data from the SW values, which enhances the data efficiency of the proposed ACNS through sensing and memorization. The accuracy of the prediction is expected to be improved with more reproducibility of devices and the use of arrayed devices. The proposed ACNS can also enhance energy efficiency by reducing the amount of data. Furthermore, it is expected that the SW values from the device can be encoded and fed into a neuromorphic hardware for practical neuromorphic signal processing.

## Discussion

Our study successfully demonstrated that the excitatory and inhibitory synaptic functions of the ACNS are similar to excitatory and inhibitory neurotransmitter release in biological synapses. An ACNS based on an OECT with chemoresponsive ionogel acting as a gate electrolyte can operate simultaneously at a low voltage and in a range of conductance states, depending on the duration, amplitude/concentration, and repetition of the applied chemical and electrical stimuli. Various synaptic properties were emulated in the ACNS by varying the electrical stimulation conditions, with high SW in the case of IG-gated devices. Interestingly, the chemical pulsing resulted in the chemically driven gating effect, which effectively modulated PSC, and unusual long-term retention and memory behavior due to the unique combination of organic semiconductor channel and chemoresponsive IG material. In our device, chemical stimulation increased the potential in pre-synapse membrane and, in turn, modulated postsynaptic current, which closely emulates the ORNs connected with projection neurons.

The simple design of the ACNS incorporates both chemical and electrical gating phenomena and can achieve a prolonged stable response even after random electrical and chemical stimulation. Previous studies have seldom reported sensing, memorization, forgetting, and overall modulation of sensory signals under simultaneous stimulation of electrical and chemical stimuli in a single device. Although we used only one type of gas as a chemical stimulus, the same approach can be used to incorporate more sensing layers and arrayed devices that sense and differentiate among multiple chemosensory signals. Modulation of the SW value, which can be varied by the different patterns of stimuli, can be used to classify different stimuli using machine learning. The SW values can also be encoded as spiking signals and supplied for bio-inspired neuromorphic signal processing, such as those involved in a spiking neural network, with significant reductions in the amount of data required. In conclusion, the biomimetic chemosensory neuronal system described here exhibits various functions similar to chemosensory neurons in the biological olfactory system. This work can further broaden the research toward a chemosensory system for use in the artificial olfactory systems.

## Methods

### Materials

For the OECT channel layer, PEDOT:PSS (poly(3,4-ethylenedioxythiophene) polystyrene sulfonate) solution was purchased from Heraeus (Clevios PH 1000). 1% volume Capstone FS-30, 4% volume ethylene glycol (EG), 0.5% volume glycidyloxypropyl)trimethoxysilane (GOPS) and 0.04 g/ml xylitol were added in PEDOT:PSS and stirred overnight before their use. Adding EG, GOPS, and xylitol to the PEDOT:PSS channel improves electrical and mechanical stability[53]. Capstone was purchased from Chemours, and EG, GOPS, and xylitol were purchased from Sigma Aldrich.

The IG was prepared by mixing the IL 1-Ethyl-3-methylimidazolium bis(trifluoromethylsulfonyl)imide ([EMIM][TFSI]), poly(ethylene glycol) diacrylate (PEGDA) monomer, and 1-hydroxycyclohexyl phenyl ketone photoinitiator (Photoiniator 184) in the ratio of 90:8:2. All three components of the IG were purchased from Sigma Aldrich. Different concentrations (90 wt.%, 60 wt.%, and 40 wt.%) of the IG were prepared and tested, out of which 90 wt.% was selected for the IG-gated device. The comparison of different concentrations of the IG is illustrated in the Supplementary information.

### OECT fabrication

The transistor was fabricated on a 50-μm-thick polyimide film, sonicated in acetone, ethanol, and deionized water. 75 nm-thick source, drain, and gate electrodes were deposited by thermal evaporation of gold through a shadow mask. The prepared PEDOT:PSS solution was then spin-coated as the channel layer between the source-drain electrodes (channel area = 5 × 1 mm) followed by annealing at 150 °C for 2 h in an $N_2$ environment has been reported to result in acceptable electrical and mechanical stability in electrolytes[53]. The source-drain electrodes were encapsulated using polydimethylsiloxane, and the devices were treated with $O_2$ plasma before the electrolyte was dropped on the channel and gate area to reduce the surface tension due to the hydrophobic nature of IL. 1 μL of IL was directly dropped onto the IL-gated OECTs. For IG-gated OECTS, the prepared IG solution was spin-coated on the channel and gate area and cured for 20 s under ultraviolet radiation (365 nm wavelength, 5 mW/m² power). The schematics of the detailed fabrication steps are shown in Supplementary Fig. S16a. The device design is shown in Supplementary Fig. S16b.

### Measurements

The responsive properties of the device toward a target gas, $NO_2$, were examined using a custom-built gas-sensing chamber connected to gas source by mass flow controllers (KRO-4000S, KMB Tech) to control parameters such as gas flow rate and humidity. All the experiments

were performed at room temperature (20 to 25 °C) and relative humidity of 20%, where $N_2$ was used as a reference and balancing gas. All electrical and time-dependent measurements of PSC were performed using a semiconductor parameter analyzer (Keithley, 4200 SCS). The channel layer thickness was examined by field-emission scanning electron microscopy (JEO JSM-6500F) (Supplementary Fig. S16c). The thickness of the IG layer was verified by a custom digital optical microscope connected to a digital camera.

## Data availability
Source data are provided with this paper.

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

## Acknowledgements

This research was supported by the Basic Science Research Program (Grant Nos. 2019R1A6A1A03033215 and 2020R1A2C3013480) through the National Research Foundation of Korea (NRF), funded by the Ministry of Science and Technology and the Ministry of Education.

## Author contributions

H.H.C. carried out the experiments, analyzed the data, and wrote the manuscript. D.H.L. developed the initial devices and fabrication processes and developed the measurement protocol. A.B. conceptualized some experiments, analyzed the data, discussed the ideas, carried out computation for machine learning, and reviewed and edited the manuscript. N.-E.L. planned and supervised the whole work and reviewed and edited the manuscript.

## Competing interests

The authors declare no competing interests.
