## [Peer Review File · Nature Communications]

REVIEWER COMMENTS

Reviewer #1 (Remarks to the Author):

The manuscript presents an artificial gas-sensing neuronal synapse to mimic excitatory and inhibitory neurotransmitter operation in the synapses of receptor neurons, projection neurons and interneurons inside the glomerulus of the olfactory bulb. The artificial synapse is made of an organic electrochemical transistor with an ion gel sensitive to changes in NO₂ concentration as excitatory chemical stimuli and a gate electrode responsive inhibitory electrical stimuli. The authors claim that the device closely emulates the synaptic connections between the olfactory neurons, that exhibits long retentive memory (hundreds of seconds), and that can be used to sense and differentiate among multiple gases in artificial olfactory systems. The study is on a topic of relevance and interest to Nature Communications readers.

The paper is well written thoroughly. However, this reviewer believes the current version of the manuscript is not ready for publication as some technical questions remain that should be clarified:

1. (lines 206 to 209, and line 219 to 220) Please provide a sweep on spike amplitude, duration and number vs. the exhibited retention times to show the transition from short-term to long-term plasticity, and the range of such times.
2. (lines 269 to 271). For a pulse of 30s, what is the limit of detection of the device? Please elaborate on its dependency on the duration of the pulses.
3. (lines 362 to 365). A fixed time of 500s does not seem to be enough for the two upper measurements of ΔPSC in Fig.4 d) and e) to stabilize. Please repeat the calculation of the normalized change of SW and the illustrations leaving instead 500s after the application of the second stimuli.
4. (lines 394 and 396). The results of the paper seem to resemble more synaptic plasticity than electrical or chemical (typically low-pass) signaling between neurons. Please clarify which specific functionality you achieve to emulate. A graphical comparison between the biological response of glomerulus synapses with that of the device here presented would be very welcome to highlight similarities and differences.

5. (lines 410 to 412) Please specify what design modifications would be needed to make the artificial synapse sensitive to other gases, and how these changes could modify the plasticity and retention behavior shown in the study.

6. (line 622) Please include the dependency on the number of pulsed chemical stimuli in Figure 3 as in Figure 2 i) to better compare between the extend of electrical and chemical regulation. Does the retention time depend on the magnitude of the stimuli as in the inhibitory case?

7. The results of Figure 4 (f) seem to indicate that one could perhaps deliver spike-timing-dependent plasticity by connecting post-synaptic spikes (generated from PSC integration) to the gate of the device and adjusting the amplitude of the excitatory and inhibitory stimuli. Can you please comment on that?

Minor: (line 20) "comprises of olfactory" -> "comprises olfactory"

Reviewer #2 (Remarks to the Author):

The manuscript describes an artificial chemosensory synapse (ACS) that mimics some of the signal processing capabilities found in the neuronal architecture of the natural olfactory system. The ACS working principle is based on an OECT operated with an ionic gel gate. The manuscript claims that the ionic gel interacts with the gas analyte NO₂ leading to some kind of "self-gating function". The consequence is a chemical sensor that integrates the sensing response. Integration of the sensor can be erased by applying electrical voltage pulses to the gate.

In my opinion the manuscript has several severe shortcomings:

1.) Device physics: The manuscript claims that the devices is based on an OECT, but the data presented describing OECT operation looks completely different from typical transfer curves found in literature. The impact of the gate voltage on drain current is extremely small and non-linear at higher voltages. Such incompatibility with existing literature data casts doubts on the reproducibility of the presented devices. In addition to this practical aspects, also the physicochemical understanding of OECT operation is described with wrong terminology. There cannot exist an effect such as "self generated potential". Potential in the ionic gel only generates as ions or electronic charges are transported across an interface. So NO₂ does not interact with the ions in the gel but

most likely interacts directly with Pedot removing electrons – thus generating hole charges. Such direct interactions between NO₂ and organic semiconductors are widely described in the literature.

2.) Materials and methods: Important fabrication steps are not described (How is the Pedot:Pss layer patterned? Etc.) or are impossible to be realized practically (drop casting of 1 uL of liquid – a drop contains minimum 20 uL). Such shortcomings put the presented findings into doubt. It seems as if more care was put into drawing appealing images than on conducting and reporting experiments.

3.) Neuromorphic architecture: This is an emerging research field and the field should maintain its focus. As written in the manuscript neuromorphic devices have the purpose to “sense, filter, compute, and memorize at the device level”. This focus comes from the need to increase autonomy of complex sensor systems and to reduce data-transfer between sensing units and central node. This ultimate purpose comes completely out of focus in the manuscript and it is completely unclear how the described artificial chemosensory system could be of use in a sensing application. In my opinion this reduces dramatically the general relevance of the manuscript. I

For these reasons I do not find the manuscript suitable for publication in Nat. Commun.. At maximum it might gain in interest if the authors focus more on the sensor application and skip the enforced analogy to neuronal architectures. The fact that they built an erasable integrating gas sensor could be interesting and of use for future applications.

Some more comments:

“ionic gel acting as a gate dielectric” - an ionic gel is a conductor, it should not be named “dielectric”.

The drawing in fig2b is misleading. The state (I) should be the one with the biggest number of hole charges (black dots), instead it is drawn as if there were no hole charges. Subsequently with the positive pulse, hole charges appear in the drawing even though the material is shown to be less conductive.

Reviewer #3 (Remarks to the Author):

Chouhdry et al. report on the development of an organic artificial chemosensory synapse. The core of this device is a PEDOT:PSS-based organic electrochemical transistor comprising an ion gel that acts as the chemoreceptive material and the gate electrolyte. The authors draw an analogy between this device and the human olfactory system, suggesting that this simple device design could be used to develop biomimetic chemosensors. The following is a list of major issues:

1. While I appreciate the self-gating response of the chemosensitive ion gel to NO₂, I argue that the device design concept cannot be generalized to other molecules of interest for developing biomimicking chemoreceptors. The authors state that the same approach can be extended to other gas/analytes incorporating more sensing layers and arrayed devices that sense and differentiate among multiple chemosensory signals. This statement is, however, too generic as the hypothetical mechanism (see comment 2 below) does not necessarily hold for other molecules. So what different sensing layers are the authors envisioning? How do they plan to sense and differentiate multiple chemosensory signals?

2. Connected to 1, the potential mechanism stated on page 13 is hypothetical and not supported by any experimental observations. The authors speculate that N₂O₄ dimers interact with the imidazole cation EMIM⁺ in the ion gel, causing the TFSI⁻ anions to penetrate the PEDOT:PSS layer. Except for the electrical characterization measurements, no other chemical analysis is reported to support this mechanism.

3. Some sections of the manuscript are redundant, especially concerning the analogy with the olfactory system, which is repeated multiple times throughout the manuscript.

4. I appreciate the long-term retention caused by the slow motion of ions in the ion gel, which I consider to be the highlight of the manuscript. I found, however, that a detection limit >200 ppm (see Fig 4C) is irrelevant for any practical application.

While this manuscript contains some interesting concepts, it fails to provide convincing evidence that the device design is general and competitive with currently existing technologies. So, I cannot recommend publication in Nature Communications.

RESPONSE TO REFEREES (NCOMMS-22-27007)

Reviewer #1 (Remarks to the Author):

The manuscript presents an artificial gas-sensing neuronal synapse to mimic excitatory and inhibitory neurotransmitter operation in the synapses of receptor neurons, projection neurons, and interneurons inside the glomerulus of the olfactory bulb. The artificial synapse is made of an organic electrochemical transistor with an ion gel sensitive to changes in NO₂ concentration as excitatory chemical stimuli and a gate electrode responsive inhibitory electrical stimuli. The authors claim that the device closely emulates the synaptic connections between the olfactory neurons, that exhibits long retentive memory (hundreds of seconds), and that can be used to sense and differentiate among multiple gases in artificial olfactory systems. The study is on a topic of relevance and interest to Nature Communications readers.

The paper is well written thoroughly. However, this reviewer believes the current version of the manuscript is not ready for publication as some technical questions remain that should be clarified:

1. (lines 206 to 209, and line 219 to 220) Please provide a sweep on spike amplitude, duration and number vs. the exhibited retention times to show the transition from short-term to long-term plasticity, and the range of such times.

Response:

We are grateful for the reviewer's comments regarding the retention times. One plausible feature of our device is that we were able to achieve long term memorization through our ACNS, since the device exhibited high retention of the state after stimulation (both electrical and chemical).

As evident from **Supplementary Fig. S4**, the IG (90 wt.% IL)-gated OECT specifically showed high retention times. Due to the limitations of our measurement setup, we could not perform measurements for extended periods of time, to show the full recovery of the device to its initial state. However, we analyzed the decay time constants instead. The decay time constant was calculated for both IL and IG (90 wt.% IL)-gated devices (for data in **Supplementary Fig. S3** and **S4**). It was observed that with increase in magnitude of electrical stimulus, the decay time constant (retention time) of IG (90 wt.% IL)-gated device demonstrated a significant increase as compared to IL-gated device.

The analyzed data is added in the Supplementary Information (**Supplementary Fig. S5**) as follows:

Fig. S5. Analyzed decay time constant (τ) for both IL and IG (90 wt.% IL)-gated devices (a) at different amplitudes (b) durations and (c) number of pulses of electrical stimuli (estimated for data in **Fig. S3** and **S4**). The theoretical decay time constant was calculated employing the following equation for exponential decay function:

$$PSC(t) = PSC_{in} + (PSC_0 - PSC_{in}) \exp\left(\frac{-t}{\tau}\right)$$

Where t is the time after the electrical stimulation, $PSC(t)$ is PSC at time t , PSC_{in} is the PSC before electrical stimulation, and PSC_0 is the PSC at time = 0. τ is the decay time constant which can give us information on the retention times. Large decay constants refer to longer retention times and slow decay of the current to its initial state.

Corresponding description was also included in the main manuscript (line 214 to 222):

“The higher retention time and slow recovery of the PSC to its initial state also contributed towards higher SW values. To demonstrate the retention of PSC state after electrical stimulation, the decay time constant (τ) of the device was analyzed by employing the exponential decay function. **Supplementary Fig. S5** shows the analyzed decay time constants for both IL and IG (90 wt.% IL)-gated devices at different magnitudes of electrical stimulus. It is established from the decay constant estimation that IG (90 wt.% IL)-gated device shows longer retention times compared to IL-gated device (up to 19×10^3 s at stimulation amplitude, $V_g = 1.5$ V) and slow decay of PSC to its initial state, generating higher SW values. Increased SW and τ values along with shift from STP to LTP was observed, by increasing the magnitude of electrical stimuli.”

2. (lines 269 to 271). For a pulse of 30s, what is the limit of detection of the device? Please elaborate on its dependency on the duration of the pulses.

Response:

We thank the reviewer for enquiring about the limit of detection of the device. For this purpose, we did a new experiment. We measured the device responses under different concentrations (*i.e.*, 10, 30, 50, and 100 ppm) of NO₂ with a fixed pulse duration of 30 s. We added the corresponding results in **Fig. S10c**, and the calculation of the limit of detection is incorporated in **Fig. S10d** in the revised **Supplementary information**.

To study the dependency on the duration of the pulses, we further did two new experiments. We measured the device response separately under 10 and 30 ppm of NO₂, respectively, with varying pulse duration (20, 30, and 60 s). We added the corresponding results as **Fig. S10e** and **S10f** in the revised **Supplementary information** as follows:

Fig. S10. (a) Time-dependent measurement of the drain current (I_d) upon exposure to increasing durations of chemical pulses at a NO_2 pulsing concentration of 400 ppm. (b) Time-dependent

measurement of I_d upon exposure to increasing NO_2 concentrations in a step-like manner from 300 to 800 ppm, with a gas pulse duration of 30 s. (c) Time-dependent response (ΔPSC) of the ACNS towards the low concentration of NO_2 (10, 30, 50 and 100 ppm) with a pulse duration of 30 s. (d) Calculation of limit of detection (LOD) from the data showing the fitted responses of the device. The theoretical LOD was calculated as: $LOD (ppm) = \frac{3 \times \text{standard deviation}}{\text{slope}} = 0.03$. Dependence of the device response (ΔPSC) on the pulse duration (20, 30 and 60 s) of (e) 10 ppm and (f) 30 ppm NO_2 pulses.

We also added a description in the revised manuscript, line no. 295 to 302, as follows:

“Concentration-dependent changes in PSC (*i.e.*, ΔPSC) of ACNS for lower concentration NO_2 (10, 30, 50, and 100 ppm) with a fixed pulse duration of 30 s (**Supplementary Fig. S10c**) was obtained and the limit of detection of the device was calculated as low as 30 ppb from the data (**Supplementary Fig. S10d**). Also, to study the dependency of ΔPSC on the pulse duration, the ΔPSC was estimated for 10 and 30 ppm of NO_2 with varying pulse duration (20, 30, and 60 s), as shown in **Supplementary Fig. S10e** and **S10f**, respectively. Both studies showed a gradual increase in the ΔPSC with increasing the concentration and duration of NO_2 pulses for a fixed duration and concentration, respectively.”

3. (lines 362 to 365). A fixed time of 500 s does not seem to be enough for the two upper measurements of ΔPSC in Fig.4 d) and e) to stabilize. Please repeat the calculation of the normalized change of SW and the illustrations leaving instead 500 s after the application of the second stimuli.

Response:

We appreciate the critical comment on the graphs regarding the stabilization of the ΔPSC , in which “a fixed time of 500 s does not seem to be enough for the two upper measurements of ΔPSC

in **Fig. 4d** and **4e** to be stabilized,” which were measured to predict the changes in the SW due to the time interval between two different types of stimulations (*i.e.*, chemical and electrical). Here, the Δ PSC was measured with the idea of parallel events happening in other synapses in the system. The information on SW is collected at the same point of time for all the events to estimate the relative effect of the time interval between two stimulations (*i.e.*, chemical and electrical) on the Δ PSC.

As per the reviewer’s suggestion, we have adjusted the timescale of the graphs in both **Fig. 4d** and **4e** to show the stabilization of Δ PSC after the application of the second stimuli. The time to stabilize the Δ PSC after the two stimulations is adjusted to be long enough for the four measurements (with varying intervals) to reach a stable state in each graph, respectively, in the modified **Fig. 4d** and **4e**, as shown below. We recalculated the SW after 100 s of two stimulations for both **Fig. 4d** and **4e** (initially at fixed time of 500 s). The recalculated values are shown in **Fig. 4f** (normalized) and **Supplementary Fig. S14** (quantitative). The modified figures are shown below:

Fig. 4. Excitatory and inhibitory functions of the ACNS. (a) Chemical synapses inside glomerulus acquire different action potentials from ORN and PGC and transfer the modulated postsynaptic signals to the olfactory bulb. (b) Schematic of the ACNS, mimicking the biological

olfactory system and showing the excitatory and inhibitory functions. The chemical stimulus contributes to excitatory PSC by diffusion of anions, whereas the electrical stimulus results in PSC inhibition by driving cations into the channel. The net modulated PSC is collected at the source-drain electrodes. (c) The ACNS with pulsed chemical stimuli at different concentrations followed by an equal number of electrical pulses, showing potentiation first and then inhibition functions. The changes in PSC with (d) excitatory followed by inhibitory and (e) inhibitory stimuli followed by an excitatory stimulus with different time intervals between the two types of stimuli. The consequent normalized SW is calculated and presented in (f). (g) Under random excitatory and inhibitory stimuli, the ACNS shows gradual modulation in the PSC according to the time and number of applied pulses.

Fig. S14. Observed SW values in Fig. 4f without normalization.

The main manuscript was modified accordingly (line 386 to 388):

“The overall normalized change in SW (measured after 100 s of two stimulations in all cases) vs. Δt (the difference in the stimulation time between the excitatory and inhibitory events) was calculated and is presented in **Fig. 4f.**”

4. (lines 394 and 396). The results of the paper seem to resemble more synaptic plasticity than electrical or chemical (typically low-pass) signaling between neurons. Please clarify which specific functionality you achieve to emulate. A graphical comparison between the biological

response of glomerulus synapses with that of the device here presented would be very welcome to highlight similarities and differences.

Response:

We realize the reviewer's concern regarding the clarity of the specific functionality that we achieve to emulate. A significant difference we understood after studying literatures on biological olfaction and the artificial emulation of olfaction is the nature of the signals in these systems. The biological neuronal systems usually comprise electrical action potentials or spikes, where the chemical changes in the stimuli and outputs are often translated into electrical spikes while transmitted through the neurons. The artificial systems, on the other hand, focus on the emulation of the overall behavior of the biological system instead of showing precisely similar responses and usually generate continuous analog outputs.^{R1}

In this work, we focused on mimicking the behavior of excitatory and inhibitory neurotransmitter release in the synapses between the neurons in the biological olfactory system. It is established that the biological olfactory system intrinsically pre-processes and modulates the neuronal signals before sending the information to the brain. It would be difficult to mimic the biological functionalities entirely. Still, we tried to develop a system that can intrinsically modulate the input signals and have functionalities such as memorization and forgetting of signals. The future prospects of this work may include using pre-processed data from the artificial system for neuromorphic architectures, reducing the need for data transducers and pre-processing elements.

Zak *et al.*^{R2} measured antagonistic activity inside the glomerulus of rats. Two different gases and their mixtures were introduced to the rats, and the change in responses from different olfactory sensory neurons in the glomerulus was observed, which showed suppression and synergy in

different cases.^{R2} We can get an idea regarding potentiation and inhibition from this study, which happens locally in the glomerulus. Our goal was to emulate a similar functionality from our work.

“Antagonism in individual sensory neurons: (a) Experimental setup for imaging OSN somata in the epithelium. (b) Example image of OSN somata and selected ROIs from one of six mice. Odor responses for two odors (Methyl tiglate and Isobutyl propionate) and a mixture of both odors from selected OSNs are shown on right as $\Delta F/F$ time courses. Red bar under traces denotes odor delivery time. Scale bar in image is 30 μm . (c) Example dose response curves from two selected OSNs. Each point is the average of 3–5 trials.”

Another example of odor-evoked inhibition and synergy can be seen in the study by Inagaki *et al.*,^{R3} where they showed that odors produce not only excitatory but also inhibitory responses in olfactory sensory neurons inside the glomerulus, depending on the binding receptors. They have summarized that “responses to odor mixtures are extensively suppressed or enhanced in olfactory sensory neurons. When high concentrations of odors are mixed, widespread antagonism suppresses the overall response amplitudes and density. In contrast, a mixture of low concentrations of odors often produces synergistic effects and boosts the faint odor inputs. Thus, odor responses are extensively tuned by inhibition, antagonism, and synergy at the most peripheral level, contributing to robust sensory representations.”^{R3}

“Odor-Evoked Inhibitory Responses in OSN Axons: (A) Two-photon calcium imaging of OSN axon terminals in vivo. OSN-GCaMP3 mice (OMP-tTA BAC Tg crossed with R26-TRE-GCaMP3 BAC Tg) were used for in vivo imaging of the OSN axon terminals in the glomerular layer. Only anesthetized mice were analyzed except for Figures S1E and S1F. Odors were delivered by a custom-made olfactometer. (B) Excitatory (red) and inhibitory (blue) responses at OSN axon terminals in the glomerular layer. Mean $\Delta F/F_0$ per pixel during the first 10 s from the stimulus onset is shown. Scale bar, 200 μm . A, anterior; P, posterior; M, medial; L, lateral. (C) Glomerular and odor specificity of excitatory and inhibitory responses. Glomeruli showing responses to at least one odorant were analyzed. Glomeruli are clustered based on the number of excitatory (4e-1e) and inhibitory (1i-4i) odors and for bidirectional (e/i) responses. Only significant responses (response mean is >3 SD above/below baseline) were categorized as excitatory or inhibitory. $n = 299$ glomeruli from 5 mice. A total of 215 glomeruli showing significant responses to at least one odorant are shown. (D) Representative glomerular (Glo) responses at OSN axon terminals to odors (diluted at 0.5%). Odors were delivered to the nose for 5 s (shown in gray) under freely breathing conditions. Excitatory and inhibitory responses were defined when the response peak and trough

were >5 SD above/below baseline, respectively. Biphasic responses demonstrated both excitatory and inhibitory responses. (E) Polarity of glomerular responses to each odor. Fractions out of the total number of glomeruli are shown. (F) Tuning specificity of excitatory and inhibitory responses. Data are from 186 and 53 odor-glomerulus pairs for excitatory and inhibitory responses, respectively. Biphasic glomeruli are categorized into both excitatory and inhibitory ones here.”

- R1. Manzini, I., Schild, D. & di Natale, C. Principles of odor coding in vertebrates and artificial chemosensory systems. *Physiol Rev* **102**, 61–154 (2022).
- R2. Zak, J. D., Reddy, G., Vergassola, M. & Murthy, V. N. Antagonistic odor interactions in olfactory sensory neurons are widespread in freely breathing mice. *Nature Communications* *2020 11:1* **11**, 1–12 (2020).
- R3. Inagaki, S., Iwata, R., Iwamoto, M. & Imai, T. Widespread Inhibition, Antagonism, and Synergy in Mouse Olfactory Sensory Neurons In Vivo. *Cell Rep* **31**, 107814 (2020).

5. (lines 410 to 412) Please specify what design modifications would be needed to make the artificial synapse sensitive to other gases, and how these changes could modify the plasticity and retention behavior shown in the study.

Response:

Ionic liquids have been used as gas-sensing/solvating materials by many researchers previously. ^{R4,R5} We believe that the gas-solvating properties of different ionic liquids to make ion gels that can be used to extend further our research towards the fabrication of an artificial synapse sensitive to various gases. Additionally, an ionic liquid can respond differently to different gases. We performed an additional experiment to show the gas sensing property of the EMIM TFSI-based IG towards four different analytes (NO₂, NH₃, SO₂, acetic acid). The device showed different responses towards NO₂ and NH₃, while we observed a small response towards SO₂ and acetic acid. The responses indicate that there is a difference in responsivity of the specific ionic liquid towards different gases. The response of EMIM TFSI towards NH₃ is opposite to and distinguishable from that of NO₂ (shown in the figure below).

We added the measured responses in the **Supplementary Fig. S8** as follows:

Fig. S8. Gas sensing property of the IG (90 wt.% IL)-gated OECT towards four different analytes of NO₂, NH₃, SO₂, and acetic acid. The device showed distinct responses towards NO₂ and NH₃, while we observed relatively small responses towards SO₂ and acetic acid.

The corresponding description was added in the main manuscript (line 252 to 254) as follows:

“To emulate the concept, our ACNS establishes a chemoreceptive function for chemical stimulation using an IG that can respond to the chemical properties of a gas. We checked the device responses under four different analytes (NO₂, NH₃, SO₂ and acetic acid) as shown in **Supplementary Fig. S8** and continued the further experiments with NO₂. Upon exposure to NO₂ gas, the gas molecules interact with cations in the IG, which drives ions into the channel in the manner of a neurotransmitter, resulting in the modulation of PSC.”

To further assess our claim of using different ionic liquids with gas-solvating properties to extend our research towards fabricating artificial synapses that can be sensitive, selective, and distinguish various gases. We fabricated an ACNS based on a different ion gel based on **BMIM TFSI** (1-butyl-3-methylimidazolium bis(trifluoromethylsulfonyl)imide). To our surprise, IG (BMIM TFSI)-gated ACNS showed a much higher response ($\Delta\text{PSC} \sim 1500 \mu\text{A}$) towards NO₂ gas

compared to IG(EMIM TFSI)-gated ACNS (shown in the graph below). This can open a new doorway towards the possibility of using multiple IGs and analytes to fabricate an array of devices that can sense and distinguish the different gases through pattern recognition.

Fig. R1. Response of BMIM TFSI based IG (90 wt.% IL)-gated OECT towards 400 ppm NO₂ ($T_{\text{on}} = 60$ s).

- R4. Lei, Z., Dai, C. & Chen, B. Gas Solubility in Ionic Liquids. *Chem. Rev.* **114**, 1289–1326 (2014).
- R5. Rehman, A. & Zeng, X. Methods and approaches of utilizing ionic liquids as gas sensing materials. *RSC Adv.* **5**, 58371–58392 (2015).

6. (line 622) Please include the dependency on the number of pulsed chemical stimuli in Figure 3 as in Figure 2i) to better compare between the extend of electrical and chemical regulation. Does the retention time depend on the magnitude of the stimuli as in the inhibitory case?

Response:

We are thankful to the reviewer for this essential comment. As per the reviewer’s suggestion, we performed new experiments and observed the response for both the IL- and IG-gated devices by varying the number of pulsed chemical stimuli. We have included the below figures to show

the dependency on the number of pulsed chemical stimuli in **Fig. 3f** and **3i** as in **Fig. 2f** and **2i** for better comparison between the extent of electrical and chemical regulation.

The revised figure and caption are as follows:

Fig. 3. Chemical regulation of the ACNS. (a) A biological chemical synapse with chemical stimulus. A chemical stimulus (in the form of gas molecules) is sensed by the ORN, and a

corresponding action potential is then transported through the neuron, releasing neurotransmitters via exocytosis into the synaptic cleft. The resulting changes in Δ PSC are transported by PN to higher olfactory centers for further processing. (b) An artificial chemical synapse with chemical stimulus. NO_2 dimerizes to form N_2O_4 in the stable state, facilitating π - π interactions between the $[\text{EMIM}]^+$ cations and N_2O_4 . Cations in the electrolyte are solvated by gas molecules, resulting in diffusion of $[\text{TFSI}]^-$ anions into the channel to interact with PEDOT^+ , increasing the Δ PSC due to gating inducement of the doping effect. (c) Transfer curves show a positive shift in threshold voltage (V_{th}) upon exposure to NO_2 gas (400 ppm). The IG (90 wt.% IL)-gated OECT shows a larger positive shift and greater gas absorption. (d) Time-dependent measurement of the Δ PSC with different concentrations (200, 400, and 600 ppm) of NO_2 with $T_{\text{on}} = 30$ s. The corresponding SW is presented in (g). (e) Time-dependent measurement of Δ PSC with different duration times, i.e., T_{on} of NO_2 (200 ppm) exposure. The corresponding SW is quantified in (h) for both IL- and IG-gated devices. (f) Time-dependent measurement of Δ PSC with different numbers of NO_2 (200 ppm) pulses, i.e., P_n (1, 2, and 5 pulses), indicates a gradual increase in SW, which is depicted in (i).

The main manuscript (lines 278 to 287) has been revised accordingly:

“The dynamic Δ PSC data of the IL- and IG (90 wt.% IL)-gated devices was measured at $V_d = -0.2\text{V}$ with different concentrations, durations, and successive periods of exposure to the chemical (NO_2 gas) pulses. The results showed an increase in Δ PSC when the concentration of the NO_2 pulse (with a duration of 30 s) was increased from 100 to 600 ppm (**Fig. 3d**). The higher the concentration, the stronger the response. Similarly, the dependence of Δ PSC on the chemical pulse duration (**Fig. 3e**) and the pulse number (**Fig. 3f**) indicates increases in the response, with increasing the duration and number, respectively. This was confirmed by the estimated SW at 500 s for the dependence of Δ PSC on the pulse concentration, duration and number. The SW value increased from 0.04% to 3% in the case of the IL-gated device and 1.2% to 16% for the IG-gated

device when the NO₂ concentration increased from 200 to 600 ppm (Fig. 3g). Similarly, a gradual increase in SW was observed upon increases of the pulse duration (Fig. 3h) and number (Fig. 3i).”

The reviewer inquired if “the retention time depend on the magnitude of the stimuli as in the inhibitory case”. Since the device showed a long-term memory behavior, which can be seen in the graphs above (Fig. 3d, 3e and 3f), the device maintained its state after exposure to gas. The synaptic weight of the device did show dependence on the magnitude of the stimuli. The calculation of retention time is challenging due to the long-term memory behavior and negligible recovery of the device. The figure below shows the extended measurement of the graph in Fig. 3d, where we can get an idea regarding the stability of the device current at a fixed state after exposure to chemical stimulus. Due to this behavior, we realized that the recovery the device to its initial state is possible by applying electrical stimulus.

Fig. R2. The figure shows the extended graph (until 1700 s) from Fig. S3a (third graph) with 600 ppm NO₂ exposed for 30 s.

7. The results of Figure 4 (f) seem to indicate that one could perhaps deliver spike-timing-dependent plasticity by connecting postsynaptic spikes (generated from PSC integration) to the gate of the device and adjusting the amplitude of the excitatory and inhibitory stimuli. Can you please comment on that?

Response:

We certainly agree with the reviewer's comment regarding the delivery of spike-timing-dependent plasticity (STDP) by connecting postsynaptic spikes (generated from PSC integration) to the gate of the device and adjusting the amplitude of the excitatory and inhibitory stimuli.

STDP is a form of learning mechanism in neurons forming synapses, which can result in the potentiation and inhibition of signals due to closely timed potentials. It plays a significant role in overall memory and learning, adaptation, in conjunction with strengthening, weakening, and fine-tuning of neuronal circuits. STDP is commonly demonstrated in the context of artificial machine learning algorithms, and spike-based computations for memristive devices.^{R6,R7,R8,R9} Emulation of the function of adaptation is possible through our ACNS by converting the PSC into postsynaptic spikes and delivering them to the gate terminal by a feedback system. We could not focus on designing such a system for our work due to the complexity of chemical and electrical presynaptic stimuli, though we had the idea of STDP in mind. To demonstrate the effect of the different action potentials' relative timing on the device's overall synaptic weight, we designed the experiment in **Fig. 4d** and **4e**. We will keep the reviewer's helpful comments and suggestions in mind for implementing neuromorphic functions in our future works.

- R6. Serrano-Gotarredona, T., Masquelier, T., Prodromakis, T., Indiveri, G. & Linares-Barranco, B. STDP and sTDP variations with memristors for spiking neuromorphic learning systems. *Front Neurosci* **0**, 2 (2013).

- R7. Yu, F. *et al.* Restickable Oxide Neuromorphic Transistors with Spike-Timing-Dependent Plasticity and Pavlovian Associative Learning Activities. *Adv Funct Mater* **28**, 1804025 (2018).
- R8. Park, M. *et al.* Neuron-Inspired Time-of-Flight Sensing via Spike-Timing-Dependent Plasticity of Artificial Synapses. *Advanced Intelligent Systems* **4**, 2100159 (2022).
- R9. Zamarreño-Ramos, C. *et al.* On spike-timing-dependent-plasticity, memristive devices, and building a self-learning visual cortex. *Front Neurosci* **0**, 26 (2011).

Minor: (line 20) "comprises of olfactory" -> "comprises olfactory"

Response:

The correction was made in the revised manuscript (line 20) as follows:

“The human olfactory system **comprises olfactory** receptor neurons, projection neurons, and interneurons that perform remarkably sophisticated functions, including sensing, filtration, memorization, and forgetting of chemical stimuli for perception.”

Reviewer #2 (Remarks to the Author):

The manuscript describes an artificial chemosensory synapse (ACS) that mimics some of the signal processing capabilities found in the neuronal architecture of the natural olfactory system. The ACS working principle is based on an OECT operated with an ionic gel gate. The manuscript claims that the ionic gel interacts with the gas analyte NO₂ leading to some kind of “self-gating function”. The consequence is a chemical sensor that integrates the sensing response. Integration of the sensor can be erased by applying electrical voltage pulses to the gate.

In my opinion the manuscript has several severe shortcomings:

1) Device physics: The manuscript claims that the devices is based on an OECT, but the data presented describing OECT operation looks completely different from typical transfer curves found in literature. The impact of the gate voltage on drain current is extremely small and non-linear at higher voltages. Such incompatibility with existing literature data casts doubts on the reproducibility of the presented devices.

Response:

We highly appreciate the reviewer’s comments regarding the device's behavior and will try to answer them in detail.

OECTs based on PEDOT:PSS as a p-type channel material have been widely reported in many literatures. ^{R10,R11,R12,R13} The reported OECT operates in the depletion mode, which means that the device is turned on in the normal state and is turned off when a positive gate voltage is applied. PEDOT:PSS channel is highly conducting in its pristine state. A positive gate biasing leads to penetration of the cations from the ion gel into the channel resulting in the depletion of holes, i.e.,

dedoping of the channel. The typical transfer curve of an OEET operating in depletion mode is shown in the figure below.

“The device physics of organic electrochemical transistors. (a) The typical structure of an organic electrochemical transistor (OEET), showing the source (S), drain (D), electrolyte and gate (G). (b) Transfer curve showing depletion-mode operation of an OEET with a conducting polymer channel. At zero gate voltage, holes on the conducting polymer contribute to a high drain current and the transistor is ON. When a gate voltage is applied, the holes are replaced by cations and the transistor is OFF. (c) Transfer curve showing accumulation-mode operation of an OEET with a semiconducting polymer channel. At zero gate voltage, the channel has few mobile holes, and the transistor is OFF. When a gate voltage is applied, holes accumulate and compensate injected anions, and the transistor is ON” ^{R10}

The following references can support this explanation in more detail:

- R10. Rivnay, J. *et al.* Organic electrochemical transistors. *Nature Reviews Materials* 2018 3:2 **3**, 1–14 (2018).
- R11. Friedlein, J. T., McLeod, R. R. & Rivnay, J. Device physics of organic electrochemical transistors. *Org Electron* **63**, 398–414 (2018).
- R12. Tu, D. & Fabiano, S. Mixed ion-electron transport in organic electrochemical transistors. *Appl. Phys. Lett.* **117**, 080501 (2020).
- R13. Inal, S., Malliaras, G.G. & Rivnay, J. Benchmarking organic mixed conductors for transistors. *Nat Commun* **8**, 1767 (2017).

Following are some examples of the typical output and transfer curves of OECT from other literature:

Output Curves	Transfer curves	Ref.
		R14
		R15
		R16

The reviewer showed concerns about the reproducibility of our devices. The transfer curves of the devices from a single batch are plotted to show the reproducibility, as displayed below:

We can see a minimal variation in the transfer curves, V_{th} , and I_{on}/I_{off} (range $\sim 10^3$) of the devices from a single batch (6 devices on the same substrate). We added this additional data in **Supplementary Fig. S2** as follows:

Fig. S2. (a, c) Transfer and (b, d) output characteristics for IG (90 wt.% IL)- and IL-gated devices. (e) Transfer characteristics of six IG (90 wt.% IL)- gated devices fabricated on a same substrate. (f) Threshold voltage (V_{th}) and (g) On-off current ratio (I_{on}/I_{off}) of the six devices shown in (e).

The main text is modified accordingly from lines 154 to 158:

“The fundamental transfer and output characteristics of IL- and IG (with 90 wt.% IL)-gated OECTs (**Supplementary Fig. S2a, S2b, S2c, and S2d**) demonstrated device operation in depletion mode. The transfer curves of six IG (90 wt.% IL)-gated devices fabricated on the same substrate show good reproducibility in their transfer curves (**Supplementary Fig. S2e**) and also their estimated V_{th} and I_{on}/I_{off} values (**Supplementary Fig. S2f and S2g**).”

The devices from different batches showed some variation, as shown in the transfer curves of our devices from different batches below. Slight variation in the data is inevitable in the case of devices based on organic materials because the devices are fabricated, stored, and measured in the ambient environment.

Fig. R3. Transfer curves of thirteen devices from different batches.

The I_{on}/I_{off} and the V_{th} values of the devices clearly indicate the effects of V_g on the drain current, which is also supported by the literature above.^{R14,R15,R16,R17,R18} The properties, such as I_{on}/I_{off} (typically in the range of 10^3 for OECTs) and V_{th} , can vary according to the device design and

materials used. Since we have used an IL/IG for electrolytic gating, instead of typically used aqueous electrolytes in OECTs, with V_{th} of less than 1V, the device turns off at slightly higher voltages. In return, we obtain better device properties, particularly for neuromorphic applications, since we get higher charge retention in the channel and slower charge diffusion back to the IL/IG after stimulation, as compared to the typical OECTs using aqueous electrolytes. This is due to the large size of ions and slower ion dynamics in IL/IG electrolytes.

- R14. Kan T., Wujian M. & Song G. Crosslinked PEDOT:PSS Organic Electrochemical Transistors on Interdigitated Electrodes with Improved Stability. *ACS Appl Poly Mat* **3**, 1436-1444 (2021).
- R15. Kirchan, A. E., et al. A PEDOT:PSS-based organic electrochemical transistor with a novel double-in-plane gate electrode for pH sensing application. *TRANSDUCERS 2017 - 19th International Conference on Solid-State Sensors, Actuators and Microsystems* 214–217 (2017).
- R16. Nguyen, T. D., Trung, T. Q., Lee, Y. & Lee, N.-E. Stretchable and Stable Electrolyte-Gated Organic Electrochemical Transistor Synapse with a Nafion Membrane for Enhanced Synaptic Properties. *Adv. Eng. Mater.* **24**, 2100918 (2022).
- R17. Tybrandt, K., Zozoulenko, I. v. & Berggren, M. Chemical potential-electric double layer coupling in conjugated polymer-polyelectrolyte blends. *Sci Adv* **3**, (2017).
- R18. Zhang, S. et al. Patterning of Stretchable Organic Electrochemical Transistors. *Chemistry of Materials* **29**, 3126–3132 (2017).

In addition to this practical aspects, also the physicochemical understanding of OECT operation is described with wrong terminology. There cannot exist an effect such as “self generated potential”. Potential in the ionic gel only generates as ions or electronic charges are transported across an interface.

Response:

We highly appreciate the reviewer’s comment on the “self-generated potential.” The potential in the ion gel is generated when ions or electronic charges are redistributed, resulting in polarization. Our interpretation of the self-gated potential was that the potential in the ion gel was generated without any external applied electrical energy (gate pulsing). We think the reviewer’s

comment is correct regarding the generation of potential in the ion gel due to the transport of charges. In light of this, we revised the manuscript and removed the term “self” from “self-gating.”

The changes are highlighted in the main text in the following lines:

Line 2: “A flexible artificial chemosensory neuronal synapse based on an organic electrochemical transistor with **chemical-driven-gating** function using a chemoresponsive ion gel.”

Line 28: “The proposed ACNS is based on a flexible organic electrochemical **transistor gated** by the potential generated by the interaction of gas molecules with ions in a chemoreceptive ion gel.”

Line 93: “The ACNS **is gated by** the potential generated by the interaction between NO₂ molecules with cations within the chemoresponsive ion gel.”

Line 125: “Due to the nature of **the gating** effect in the ion gel under chemical stimulation, potentiation of the OECT synapse can be obtained by chemical pulsing without accompanying electrical gate pulsing.”

Line 326: “The potentiation mechanism by chemical stimulation can therefore be explained by the **chemically driven gating effect**, which is induced by the potential in the IG generated by interactions among NO₂ molecules and cations.”

Line 331: “Inhibition of PSC can be achieved by applying an external electrical field, which can mitigate the **chemically driven gating effect**, resulting in extraction of anions in the channel and injection of cations freed from the solvated complex of NO₂ gas molecules and cations.”

Line 442: “Interestingly, the chemical pulsing resulted in the **chemically driven gating effect**, which effectively modulated PSC, and unusual long-term retention and memory behavior due to the unique combination of organic semiconductor channel and chemoresponsive IG material.”

Line 673: “Cations in the electrolyte are solvated by gas molecules, resulting in diffusion of [TFSI]⁻ anions into the channel to interact with PEDOT⁺, increasing the PSC due to **gating inducement** of the doping effect.”

Line 3 (Supplementary information): “A flexible artificial chemosensory neuronal synapse based on an organic electrochemical transistor with **chemical-driven-gating** function using a chemoresponsive ion gel.”

So NO₂ does not interact with the ions in the gel but most likely interacts directly with Pedot removing electrons – thus generating hole charges. Such direct interactions between NO₂ and organic semiconductors are widely described in the literature.

Response:

The reviewer’s comment is appreciated regarding their concern that NO₂ directly interacts with the PEDOT:PSS, removing electrons from the channel layer, instead of interacting with the IL/IG. We fabricated a device without any IL/IG electrolyte and compared the response with an IG-gated OECT (graph shown below). Compared to the IG-gated OECT, the device without any IG showed a small, recoverable response to 400 ppm NO₂. The low response shows a reversible interaction between NO₂ and PEDOT:PSS. Our goal of emulation of long-term memorization can only be fulfilled by utilizing the IL/IG-gated OECT structure. The interactions between gases and ILs have been studied by many researchers in the past, which is advantageous for us to use it as a gas sensing as well as electrolytic gating layer.^{R19,R20,R21}

We added this data in **Supplementary Fig. S9** as follows:

Fig. S9. (a) Transfer characteristics of IG (90 wt.% IL)-gated OECT after exposure to increasing concentrations of NO₂ (100–500 ppm) with an exposure time of 30 s. (b) Comparison of the responses of the IG (90 wt.% IL)-gated OECT and an OECT without any IG.

The description was also added in the revised manuscript (line 269 to 274) as follows:

“We also compared the responses of an IG (90 wt.% IL)-gated OECT and an OECT without IG (**Supplementary Fig. S9b**). Compared to the IG (90 wt.% IL)-gated OECT, the device without IG showed a small, recoverable response to 400 ppm NO₂. The results indicate that the IG (90 wt.% IL)-gated OECT structure can be utilized to get a high response with high retention of the induced changes in PSC via chemical-gating effect.”

- R19. Lei, Z., Dai, C. & Chen, B. Gas Solubility in Ionic Liquids. *Chem. Rev.* **114**, 1289–1326 (2014).
- R20. Rehman, A. & Zeng, X. Methods and approaches of utilizing ionic liquids as gas sensing materials. *RSC Adv.* **5**, 58371–58392 (2015).
- R21. Buzzeo, M. C., Hardacre, C. & Compton, R. G. Use of room temperature ionic liquids in gas sensor design. *Anal Chem* **76**, 4583–4588 (2004).

2.) Materials and methods: Important fabrication steps are not described (How is the PEDOT:PSS layer patterned? Etc.) or are impossible to be realized practically (drop casting of 1 μ L of liquid – a drop contains minimum 20 μ L). Such shortcomings put the presented findings into doubt. It seems as if more care was put into drawing appealing images than on conducting and reporting experiments.

Response:

As the reviewer commented, the fabrication steps are added in the **Supplementary Figure S16a**, as follows:

Fig. S16. (a) Schematic of the fabrication steps. (b) Photograph of the fabricated device and (c) cross-sectional FE-SEM image of PEDOT:PSS channel layer (thickness of approximately 143 nm).

The changes in the main manuscript lines: (486 to 488) are as follows:

“The schematics of the detailed fabrication steps are shown in **Supplementary Fig. S16a**. The device design is shown in **Supplementary Fig. S16b**.”

The comment regarding the “drop casting of 1 μL of liquid whereas a drop contains minimum 20 μL ” is appreciated. The 1 μL droplet was dropped using a micropipette (Dragonlab) as shown below:

Fig. R4. The first figure shows the micropipette (from Dragonlab) used in our experiment. The next two figures show the dropping of IL/IG on the device using the micropipette.

We further made changes in the following lines of the revised manuscript and write “dropped” instead of “drop-casted” since it might confuse the readers.

Line 228: “The dropped IL exhibited a stability problem, presumably due to hydrophobicity, instability due to the flow of IL on the device surface, and difficulty establishing a uniformly thin layer.”

Line 483: “The source-drain electrodes were encapsulated using polydimethylsiloxane, and the devices were treated with O₂ plasma before the electrolyte was dropped on the channel and gate area to reduce the surface tension due to the hydrophobic nature of IL.”

Line 484: “1 μ L of IL was directly dropped onto the IL-gated OECTs.”

3.) Neuromorphic architecture: This is an emerging research field and the field should maintain its focus. As written in the manuscript neuromorphic devices have the purpose to “sense, filter,

compute, and memorize at the device level”. This focus comes from the need to increase autonomy of complex sensor systems and to reduce data-transfer between sensing units and central node. This ultimate purpose comes completely out of focus in the manuscript and it is completely unclear how the described artificial chemosensory system could be of use in a sensing application.

Response:

We understand the reviewer’s concerns regarding the applications to neuromorphic engineering aspects of this work. The emerging work on artificial neuromorphic perception systems is expected to be able to reduce the data amount and simplify the hardware architectures for sensor signal processing by using the sensors which can sense, memorize and transduce the signals within a single system, with neither separate sensing and transduction units nor the complex hardware systems for emulating the biological sensory organs. The synaptic weight (SW) changes in the ACNS can be directly utilized in software-based computation techniques for training and reading of different signals, predicting the properties of applied stimuli, or can be used as the pre-processed data which are fed into neuromorphic computing hardware.

To demonstrate the capability of the ACNS for data-efficient bio-inspired perception of input stimuli, the SW values from the time-dependent response (Δ PSC) of IG (90 wt.% IL)-gated OECT under chemical stimulation (NO_2 with $T_{\text{on}} = 30$ s) with varying the concentration from 100 to 400 ppm (**Supplementary Fig. S15a**) was used for training and prediction of the concentration using a machine learning (ML) algorithm. Instead of utilizing a full time-dependent variation of Δ PSC of the device for ML, as an example, we considered the SW values at 3 instances (at 90, 210, and 330 s at equally spaced time interval of ~ 120 s) after the application of chemical stimulation (NO_2 with $T_{\text{on}} = 30$ s) for generating the dataset for ML algorithm. By using SW data, we could drastically reduce the data amount utilized for the ML algorithm.

Here, for that purpose, we have measured the responses of the ACNS eight times for each concentration (total of 32 samples) to construct a simple ML algorithm for the classification of different concentrations of NO₂ (ranging from 100 to 400 ppm) utilizing the Classification Learner application in the ML toolbox available in the MATLAB software. Before training, all the samples were shuffled randomly, and 50% (i.e., 16 samples) were used as training datasets, and the rest 50% was used for testing the performance of the ML model. Among all the available algorithms in the ML toolbox, the support vector machine (SVM) was shown to have better training and test accuracy of 87.5% and 81.2% (**Supplementary Fig. S15b**), respectively. From the result, it can be concluded that four different concentrations of NO₂ can be classified successfully after training, even utilizing significantly fewer training and test samples. Out of 16 training samples, 14 samples could be predicted accurately, whereas 3 samples are misclassified out of 16 test samples for the testing set. The test accuracy of the artificial chemosensory system can further be improved by utilizing an ACNS array. The development of artificial chemosensory systems can be useful in solving the existing huge energy consumption problem by reducing the amount of data.

Similarly, we can also determine the data classification and prediction for other device properties, such as dependence on pulse duration or number of applied chemical stimuli. Since the paper's focus is not on the direction of neuromorphic signal processing and can divert the readers' attention from the aim of this work, as per the reviewer's suggestion, we have shown a simple demonstration as an example. The measured and computed data are added as Fig. S15 in the revised Supplementary Information.

Fig. S15. (a) Time-dependent response (ΔPSC) of IG (90 wt.% IL)-gated OECT under chemical stimulation (NO_2 with $T_{on} = 30$ s) with increasing the concentration from 100 to 400 ppm. Repeated measurements were performed to get eight concentration-dependent responses for each case. (b) Normalized confusion matrix showing the classification performance (test accuracy 81.2%) of a single device to 4 different concentrations of NO_2 gas (100, 200, 300, and 400 ppm) after training with a ML model (SVM) in Classification Learner. The colored scale bar shows the test accuracy of the classification.

A description was added in the main manuscript (line 419 to 438) as follows:

“To demonstrate the capability of the ACNS for an application to the data-efficient bio-inspired perception of input stimuli, the SW values from the time-dependent response (ΔPSC) of IG (90 wt.% IL)-gated OECT under chemical stimulation (NO_2 with $T_{on} = 30$ s) with varying the concentration from 100 to 400 ppm (**Supplementary Fig. S15a**) were used for training and prediction of the concentration using a machine learning (ML) algorithm. Here, for the purpose, we measured the performance of the ACNS eight times at each concentration (total of 32 samples)

for the classification of different concentrations of NO₂ (ranging from 100 to 400 ppm) utilizing the support vector machine (SVM) in Classification Learner application in the ML toolbox available in the MATLAB software. Instead of utilizing a full time-dependent variation of Δ PSC of the device for ML, as an example, we considered the SW values at 3 instances (at 90, 210, and 330 s at equally spaced time interval of \sim 120 s) after the chemical stimulation (NO₂ with $T_{\text{on}} = 30$ s) for generating the dataset for ML. The results were shown to have training and test accuracy of 81.2% (**Supplementary Fig. S15b**). From the results, it can be concluded that four different concentrations of NO₂ can be classified successfully after training, even utilizing significantly fewer training and test data from the SW values, which enhances the data-efficiency of the proposed ACNS through sensing and memorization. The accuracy of the prediction is expected to be improved with more reproducibility of devices and the use of arrayed devices. The proposed ACNS can also enhance the energy efficiency by reducing the amount of data. Furthermore, it is expected that the SW values from the device can be encoded and fed into a neuromorphic hardware for practical neuromorphic signal processing.”

“ionic gel acting as a gate dielectric” - an ionic gel is a conductor, it should not be named “dielectric”.

Response:

We acknowledge the reviewer’s comments. We will adjust the manuscript and replace the term “gate-dielectric” with “gate-electrolyte.” The changes in the revised manuscript are highlighted in the following lines:

Line 32: “Long-term memorization of the excitatory chemical stimulus can be also erased by applying an inhibitory electrical stimulus due to ion dynamics in the chemoresponsive ion-gel gate electrolyte.”

Line 78: “Our ACNS is based on an organic electrochemical transistor (OECT) with a poly(3,4-ethylenedioxythiophene) polystyrene sulfonate (PEDOT:PSS) channel and an ion gel (IG) based on 1-ethyl-3-methylimidazolium bis(trifluoromethyl sulfonyl)imide ([EMIM][TFSI]) ionic liquid (IL), poly(ethylene glycol) diacrylate (PEGDA) monomer, and a 1-hydroxycyclohexyl phenyl ketone photoinitiator as a chemoreceptive material as well as gate electrolyte layer.”

Line 437: “An ACNS based on an OECT with chemoresponsive ion gel acting as a gate electrolyte can operate simultaneously at a low voltage and in a range of conductance states, depending on the duration, amplitude/concentration, and repetition of the applied chemical and electrical stimuli.”

We would like to include the following references ^{R22,R23,R24} in this answer to provide evidence that the “ion gel” can be alternatively named as a “gate dielectric”. Numerous studies can be found in the literature where people use the general term “dielectric” for different kinds of gating materials in transistor structures, including electrolytes.

- R22. Lee, J. *et al.* Ion gel-gated polymer thin-film transistors: Operating mechanism and characterization of gate dielectric capacitance, switching speed, and stability. *Journal of Physical Chemistry C* **113**, 8972–8981 (2009).
- R23. Cho, J. H. *et al.* Printable ion-gel gate dielectrics for low-voltage polymer thin-film transistors on plastic. *Nature Materials* 2008 7:11 **7**, 900–906 (2008).
- R24. Kim, B. J. *et al.* High-performance flexible graphene field effect transistors with ion gel gate dielectrics. *Nano Lett* **10**, 3464–3466 (2010).

The drawing in fig2b is misleading. The state (I) should be the one with the biggest number of hole charges (black dots), instead it is drawn as if there were no hole charges. Subsequently with the

positive pulse, hole charges appear in the drawing even though the material is shown to be less conductive.

Response:

We have considered the reviewer's comment and made the changes in **Fig. 2b** accordingly. The modified figure is as follows:

Fig. 2. Electrical regulation of the ACNS. (b) Schematic diagram of the artificial chemical synapse. The inset shows the operating mechanism of the artificial synapse, (I) before an action potential (V_g pulse) is applied, and (II) when an action potential ($V_g > 0$) is applied to the presynaptic gate electrode, the ions diffuse from the electrolyte to the channel which results in

change in ΔPSC , which can be observed between source and drain electrodes. (III) Ions slowly diffuse to their original distribution once the action potential is removed.

Reviewer #3 (Remarks to the Author):

Chouhdry et al. report on the development of an organic artificial chemosensory synapse. The core of this device is a PEDOT:PSS-based organic electrochemical transistor comprising an ion gel that acts as the chemoreceptive material and the gate electrolyte. The authors draw an analogy between this device and the human olfactory system, suggesting that this simple device design could be used to develop biomimetic chemosensors. The following is a list of major issues:

1. While I appreciate the self-gating response of the chemosensitive ion gel to NO₂, I argue that the device design concept cannot be generalized to other molecules of interest for developing biomimicking chemoreceptors. The authors state that the same approach can be extended to other gas/analytes incorporating more sensing layers and arrayed devices that sense and differentiate among multiple chemosensory signals. This statement is, however, too generic as the hypothetical mechanism (see comment 2 below) does not necessarily hold for other molecules. So what different sensing layers are the authors envisioning? How do they plan to sense and differentiate multiple chemosensory signals?

Response:

We appreciate the reviewer's comments regarding this work's extension for sensing other molecules of interest for developing biomimicking chemoreceptors.

We believe that the gas-solvating properties of different ionic liquids can be used to extend further our research toward the fabrication of an artificial synapse that can be sensitive to various gases. Additionally, an ionic liquid can respond differently to different gases ^{R25,R26}. Since this is a proof of concept, extensive research needs to be performed to fully develop the idea of an artificial chemosensory synapse, which can sense, differentiate and memorize multiple analytes.

We performed an experiment to show the gas sensing property of the EMIM TFSI based IG towards four different analytes (NO₂, NH₃, SO₂, acetic acid). The device showed clearly different responses towards NO₂ and NH₃, while we observed a small response towards SO₂ and acetic acid. This shows that there is a selectivity of the specific ionic liquid toward different gases. The response of EMIM TFSI towards NH₃ is opposite and distinguishable from that of NO₂ (shown in the figure below).

We added the measured responses in the **Supplementary Fig. S8** as follows:

Fig. S8. Gas sensing property of the IG (90 wt.% IL)-gated OECT towards four different analytes of NO₂, NH₃, SO₂, and acetic acid. The device showed distinct responses towards NO₂ and NH₃, while we observed relatively small responses towards SO₂ and acetic acid.

The corresponding description was added in the main manuscript (line 252 to 254) as follows:

“To emulate the concept, our ACNS establishes a chemoreceptive function for chemical stimulation using an IG that can respond to the chemical properties of a gas. We checked the device responses under four different analytes (NO₂, NH₃, SO₂ and acetic acid) as shown in **Supplementary Fig. S8** and continued the further experiments with NO₂. Upon exposure to NO₂

gas, the gas molecules interact with cations in the IG, which drives ions into the channel in the manner of a neurotransmitter, resulting in the modulation of PSC.”

We agree with the reviewer about the concern regarding sensing mechanisms for different gases, which cannot be generalized, and each gas should have a specific reaction/solvation property with different ILs (either with cation or anion). Many researchers have previously used Ionic liquids as gas-sensing/solvating materials, and various sensing mechanisms have been proposed regarding different gases.^{R25,R26,R27} We are planning to investigate different sensing mechanisms for different gases and sensing layers for our future work, which can open the gates for endless possibilities in artificial chemosensory applications. To further assess our claim of using different ionic liquids with gas-solvating properties to extend our research towards the fabrication of artificial synapses that can be sensitive, selective and distinguish diverse gases, we tested an ACNS based on a different ion gel based on **BMIM TFSI** (1-Butyl-3-methylimidazolium bis(trifluoromethylsulfonyl)imide). To our surprise, IG (BMIM TFSI)-gated ACNS showed a much higher response ($\Delta PSC \sim 1500 \mu A$) towards NO_2 as compared to IG (EMIM TFSI)-gated ACNS ($\Delta PSC \sim 100 \mu A$) (shown in the graph below). This demonstrates the possibility of using multiple IGs and analytes to fabricate an array of devices that can sense, distinguish and memorize different gases.

Fig. R5. Response of BMIM TFSI based IG (90 wt.% IL)-gated OECT towards 400 ppm NO₂ (T_{on} = 60 s).

- R25. Lei, Z., Dai, C. & Chen, B. Gas Solubility in Ionic Liquids. *Chem. Rev.* **114**, 1289–1326 (2014).
- R26. Rehman, A. & Zeng, X. Methods and approaches of utilizing ionic liquids as gas sensing materials. *RSC Adv.* **5**, 58371–58392 (2015).
- R27. Buzzeo, M. C., Hardacre, C. & Compton, R. G. Use of room temperature ionic liquids in gas sensor design. *Anal Chem* **76**, 4583–4588 (2004).

2. Connected to 1, the potential mechanism stated on page 13 is hypothetical and not supported by any experimental observations. The authors speculate that N₂O₄ dimers interact with the imidazole cation EMIM⁺ in the ion gel, causing the TFSI⁻ anions to penetrate the PEDOT:PSS layer. Except for the electrical characterization measurements, no other chemical analysis is reported to support this mechanism.

Response:

We appreciate the reviewer’s comments on the potential mechanism stated on page 13, as hypothetical and not supported by any experimental observations. We have stated that N₂O₄ dimers

interact with the imidazole cation EMIM^+ in the ion gel, causing the TFSI^- anions to penetrate the PEDOT:PSS layer.

The mechanism for NO_2 and $\text{EMIM}^+ \text{TFSI}^-$ interaction has been stated by other researchers, using the theoretical and statistical calculations, such as density functional theory (DFT) calculation performed by Jin *et al.*^{R28} in “Scalable Superior Chemical Sensing Performance of Stretchable Ionotronic Skin via a π -Hole Receptor Effect. *Adv. Mater.* **33**, 2007605 (2021)” as shown below.

“Sensing mechanism and the possible interaction energy of gases with configurations on the $[\text{EMIM}]^+[\text{TFSI}]^-$ ionic liquid using density functional theory (DFT) calculations. (a) Schematic

illustration of the working mechanism for the IGS. The dashed lines indicate bounded ions due to the ion–ion interaction, and the dashed circles indicate the dissociation of paired ions due to the ion–gas interaction. (b) Molecular electrostatic potential surfaces for NO₂, toluene, N₂O₄, [TFSI][−], [EMIM]⁺, and [EMIM]⁺[TFSI][−], where the red and blue colors denote the regions of more negative and more positive charges, respectively. The isodensity contour is 0.0004 electrons per bohr. (c) Optimized structures and binding energies of the most stable interaction modes for [EMIM]⁺[TFSI][−]–NO₂, [EMIM]⁺[TFSI][−]–toluene, and [EMIM]⁺[TFSI][−]–N₂O₄ at the M06-2X/6-31+G** level.” R28

We took advantage of the existing literature to explain our work. Performing full analysis with methods such as DFT calculation is beyond the scope of our work and could divert the reader’s attention. We will keep the reviewer’s valuable comments in mind while reporting our next work.

Below we have shown the response of PEDOT:PSS based OECT under negative and positive gate biasing. Under the negative biasing, the anions penetrate inside the channel increasing the on-state current of the device. We can find this behavior similar to the chemical gating, except for the long retention times in chemical gating. This indicates that the negative potential generated inside the ion gel due to chemical gating is indeed causing the anions to penetrate the channel. The long retention time (in the case of chemical stimuli) indicates a strong interaction between the cations and gas analytes in the ion gel, impeding the recovery of the anions in the channel to their initial state.

We added the following figure in the Supplementary information:

Fig. S11. Comparison of the device responses (Δ PSC) under positive and negative electrical gate pulsing showing opposite behaviors. The positive gate bias pulsing implies the penetration of cations while the negative gate bias pulsing implies the penetration of anions into the channel.

The description was also added in the main manuscript (line 318 to 325) as follows:

“The phenomenon of penetration of anions into the channel and an increase in the Δ PSC can be also observed by applying a negative electrical pulsing at the gate (**Supplementary Fig. S11**). The results indicate that the anions indeed diffuse into the channel upon chemical stimulation of the device. Unlike the electrical stimulation, the anions are retained in the channel for an extended period upon chemical stimulation, indicating a strong interaction between the solvated complex of cations and NO_2 in the IL/IG, hindering the diffusion of the anions in the channel to their initial positions.”

- R28. Jin, M. L. *et al.* Scalable Superior Chemical Sensing Performance of Stretchable Ionotronic Skin via a π -Hole Receptor Effect. *Advanced Materials* **33**, 2007605 (2021).

3. Some sections of the manuscript are redundant, especially concerning the analogy with the olfactory system, which is repeated multiple times throughout the manuscript.

Response:

We realize the reviewer's observation regarding the redundancy of the olfactory system analogy. We improved the manuscript accordingly, and the modifications are stated as follows:

Line 63 to 68: “The olfactory system consists of a network of chemical synapses in the glomerulus of the olfactory bulb.³¹ ~~The system receives signals from the axons of olfactory receptor neurons (ORNs) through a presynaptic membrane. Chemo-sensitive cilia express odorant receptor proteins that bind in response to different odors, and the subsequent release of neurotransmitters to the dendrites of two projection neurons (PNs), i.e., mitral cells and tufted cells (through a postsynaptic membrane) is regulated accordingly.~~^{31,32} Synaptic events in these synapses synaptic cleft can be excitatory or inhibitory. The olfactory bulb in particular has more inhibitory synapses compared with the other areas of the brain. It is composed of interneurons of periglomerular cells (PGCs) and granule cells (GCs) and innervated with excitatory neurons, which are believed to be helpful in adaptation, perception, discrimination, regulation, and attenuation of odor signals, and odor coding (e.g., shaping the overall output of the olfactory bulb).^{33–35}”

Line 103 to 105: “Biological synapses in the olfactory system form between the axons of olfactory receptor neurons (ORNs) and the dendrites of two projection neurons (PNs), i.e., mitral and tufted cells, in a spherical structure called the glomerulus, which is located in the olfactory bulb.”

Line 107 to 108: “Two inhibitory interneurons of periglomerular cells (PGCs) and granule cells (GCs) form synapses inside the glomerulus with PNs (Fig. 1a).”

Line 333 to 336: “The schematic representation of our ACNS mimicking of the functions of a chemical synapse with excitatory or inhibitory presynaptic membranes in the biological olfactory system (**Fig. 4a**), is shown in **Fig. 4b**, describing the working principle when chemically exciting and electrically inhibiting stimuli are applied.”

4. I appreciate the long-term retention caused by the slow motion of ions in the ion gel, which I consider to be the highlight of the manuscript. I found, however, that a detection limit >200 ppm (see Fig 4C) is irrelevant for any practical application.

Response:

We appreciate the reviewer’s comment on the detection limit of the device for practical applications. Since this work is a proof of concept for artificial chemosensory applications, we did not focus much on the detailed experiments at lower concentrations. Considering this comment, we would like to add the measurements below to show that the device can detect NO₂ at lower concentrations. We measured the data and added the results in **Supplementary Fig. S10**, as follows:

Fig. S10. (a) Time-dependent measurement of the drain current (I_d) upon exposure to increasing durations of chemical pulses at a NO_2 pulsing concentration of 400 ppm. (b) Time-dependent

measurement of I_d upon exposure to increasing NO₂ concentrations in a step-like manner from 300 to 800 ppm, with a gas pulse duration of 30 s. . (c) Time-dependent response (Δ PSC) of the ACNS towards the low concentration of NO₂ (10, 30, 50 and 100 ppm) with a pulse duration of 30 s. (d) Calculation of limit of detection (LOD) from the data showing the fitted responses of the device. The theoretical LOD was calculated as: $LOD (ppm) = \frac{3 \times \text{standard deviation}}{\text{slope}} = 0.03$. Dependence of the device response (Δ PSC) on the pulse duration (20, 30 and 60 s) of (e) 10 ppm and (f) 30 ppm NO₂ pulses.

We also added a description in the revised manuscript, line no. 295 to 302, as follows:

“Concentration-dependent changes in PSC (*i.e.*, Δ PSC) of ACNS for lower concentration NO₂ (10, 30, 50, and 100 ppm) with a fixed pulse duration of 30 s (**Supplementary Fig. S10c**) was obtained and the limit of detection of the device was calculated as low as 30 ppb from the data (**Supplementary Fig. S10d**). Also, to study the dependency of Δ PSC on the pulse duration, the Δ PSC was estimated for 10 and 30 ppm of NO₂ with varying pulse duration (20, 30, and 60 s), as shown in **Supplementary Fig. S10e** and **S10f**, respectively. Both studies showed a gradual increase in the Δ PSC with increasing the concentration and duration of NO₂ pulses for a fixed duration and concentration, respectively.”

REVIEWER COMMENTS

Reviewer #1 (Remarks to the Author):

Thank you for the revision. This reviewer would appreciate further clarification on the following points of the new version of the manuscript:

1. (lines 214 to 222) In my judgement, the transition from short-term to long-term plasticity is still insufficiently proven in the paper. The expression employed to calculate the theoretical time constant might not be valid for e.g. Fig S.4. This Figure shows a descending exponential drift along the first 50s that would explain why the PSCs quickly stabilize to increasingly lower values afterwards. Have you tried fitting the data accordingly to this phenomenon? Please comment on what could cause the IG (90 wt.% IL)-gated device to exhibit such an exponential trend.

2. (lines 295 to 302) Please elaborate on the procedure used to compute the SD and the slope. To calculate the SD, why did you restrict the measurements to those made from seconds 20 to 31? What is the average value of ΔPSC_0 under null NO₂ stimuli (i.e. ΔPSC_0)?

Please plot the x-axis of Fig. S10.(d)(upper) in linear scale to avoid confusion. The distance between 50 and 100 ppm should be proportional to the one between 10 and 30 (or between 30 and 50 ppm), not the same.

The calculation of the LOD should take into account ΔPSC_0 and the y-intercept, following the equation $LOD = (\Delta PSC_0 + 3 \cdot SD - y\text{-intercept}) / \text{slope}$. From Fig. S10.(d), the y-intercept seems to be located around 0.6 μA . Discarding saturation at 100 ppm, this same figure hints that the slope between 10 and 50 ppm would be approx. 0.075 $\mu A/ppm$. How did you determine the 1.55625e-6 specified in the document? With these two values - and considering a negligible ΔPSC_0 - the LOD would be approximately 8.5 ppm. Please justify if you believe otherwise.

3. (Fig. 4) It seems from (d) that the two upper signals are in the transient regime. If possible, it would be interesting to compute the τ and the initial and final ΔPSC values by fitting an exponential equation like the one used in Fig. S5. Fig.4(f) could then be obtained letting the signals stabilize to $3 \cdot \tau$ or $4 \cdot \tau$ and evaluating all SWnorm points at these same multiples of τ . Please plot all green, yellow, and red lines in Fig.4(d) and (e) up to the end of the x-axis.

Reviewer #2 (Remarks to the Author):

The authors took great effort to respond to all critiques given by the reviewers. Overall the manuscript has gained significantly in clarity and the claims are better supported by the data. In my opinion no further revisions are necessary and the manuscript is suitable for publication.

Reviewer #3 (Remarks to the Author):

The authors have satisfactorily addressed the reviewers' concerns and made substantial changes to the manuscript. Although some points concerning the detection mechanism and its generality remain unresolved, I believe this work will interest the scientific community and inspire future studies. For this reason, I suggest acceptance of this manuscript in Nature Communications.

RESPONSE TO REFEREES (NCOMMS-22-27007A)

Reviewer #1 (Remarks to the Author):

1. (lines 214 to 222) In my judgement, the transition from short-term to long-term plasticity is still insufficiently proven in the paper. The expression employed to calculate the theoretical time constant might not be valid for e.g. Fig S.4. This Figure shows a descending exponential drift along the first 50s that would explain why the PSCs quickly stabilize to increasingly lower values afterwards. Have you tried fitting the data accordingly to this phenomenon? Please comment on what could cause the IG (90 wt.% IL)-gated device to exhibit such an exponential trend.

Response:

We are grateful for the reviewer's concerns regarding the transition from short-term plasticity to long-term plasticity in this work. We performed an additional experiment to show the change from short-term plasticity (STP) to long-term plasticity (LTP) for both IL- and IG (90 wt.% IL)-gated devices. The conductance states of both devices can be modulated by changing the inhibitory electrical stimulus amplitude (V_g), duration (T_{on}), and the number (P_n) of electrical stimuli to emulate the transition from STP to LTP, as shown in **Fig. R1** and **R2**.

Fig. **R1a** and **R2a** show the typical STP to LTP transition by increasing V_g pulse amplitude (0.5, 0.8, 1.0, 1.2, and 1.5 V) while keeping the pulse width ($T_{on} = 0.5$ s) and number ($P_n = 1$) constant. Depending on the amplitude of applied positive gate voltage (V_g), the number of cations pushed into the PEDOT:PSS channel can be varied, thus channel conductance can be modulated by depleting the holes in the channel. After releasing the electrical stimulus, the cations slowly diffuse back to the electrolyte. The PSC decays to its initial state rapidly or slowly, mainly depending on

the previously applied V_g . As the amplitude of V_g increased from 0.5 V to 1.5 V, both devices transitioned from STP to LTP. The IG (90 wt.% IL)-gated device (Fig. R2a) showed a more significant change in PSC (*i.e.*, Δ PSC) than that of the IL-gated device (Fig. R1a).

Fig. R1. Time-dependent measurement of postsynaptic current change (Δ PSC) with varying (a) amplitude, duration, and the number of electrical pulses for the IL (90 wt.% IL)-gated device.

Fig. R2. Time-dependent measurement of postsynaptic current change (ΔPSC) with varying (a) amplitude, duration, and the number of electrical pulses for the IG (90 wt.% IL)-gated device.

Similarly, the STP behavior of both devices can be converted to LTP as the electrical stimuli pulse width (T_{on}) increases from 0.1 s to 3.0 s while keeping the pulse amplitude ($V_g = 1.0$ V) and number ($P_n = 1$) constant, as shown in Fig. R1b and R2b. By strengthening repetitive practice (*i.e.*, repetitive stimulation), the synaptic behavior of both devices can also be adjusted from STP to LTP. As shown in Fig. R1c and R2c, the transition from STP to LTP is evident for both devices with the increasing number (P_n) of electrical stimuli ($V_g = 1.0$ V, $T_{\text{on}} = 0.5$ s).

The reviewer showed some concern that **Supplementary Fig. S4** shows a descending exponential drift along the first few seconds of the graph (~ 20 s, as shown in **Fig. R1a**). When calculating the SW, we noted both the PSC values just before the application of the electrical stimulation (*i.e.*, PSC_{in}) and 30 s after the termination of electrical stimuli (*i.e.*, PSC_{sw} , as described in **Fig. 2c**) and further calculated the change ($|(PSC_{sw} - PSC_{in})/PSC_{in}|$) according to that point. Thus the SW values would not be affected by this initial decrease in the trend, which is due to the stabilization of the device state under no gate bias probably caused by the re-arrangement of the charge carriers, inducing a change in the drain current of the device. Therefore, we did not take this into account while fitting the data. The initial descending trend lasts for a short period and does not affect the overall change in the current (and consecutively SW) due to the electrical stimulation if the gate voltage (V_g) is small (as shown in **Fig. R3a**). At large V_g (**Fig. R3b**), the change in the current and SW is entirely proportional to the applied V_g .

Fig. R3. Enlarged graphs from **Supplementary Fig. S4a** at (a) $V_g = 0.5$ V and (b) $V_g = 1.2$ V.

As discussed in the main manuscript before, the ion dynamics in the IG-gated device are slower than that of the IL-gated device, due to the presence of the polymer in the IG. As a result of this,

the IG-gated device showed a descending trend longer than that for an IL-gated device. The IL-gated device also shows a similar trend for a very short time period at the beginning of measurement in the graphs and reaches a stable state more quickly compared to the IG-gated device.

2. (lines 295 to 302) Please elaborate on the procedure used to compute the SD and the slope. To calculate the SD, why did you restrict the measurements to those made from seconds 20 to 31? What is the average value of ΔPSC under null NO_2 stimuli (i.e. ΔPSC_0)?

Please plot the x-axis of Fig. S10.(d)(upper) in linear scale to avoid confusion. The distance between 50 and 100 ppm should be proportional to the one between 10 and 30 (or between 30 and 50 ppm), not the same.

The calculation of the LOD should take into account ΔPSC_0 and the y-intercept, following the equation $LOD = (\Delta PSC_0 + 3 * SD - y\text{-intercept}) / \text{slope}$. From Fig. S10.(d), the y-intercept seems to be located around 0.6 μA . Discarding saturation at 100 ppm, this same figure hints that the slope between 10 and 50 ppm would be approx. 0.075 $\mu A/ppm$. How did you determine the $1.55625e-6$ specified in the document? With these two values - and considering a negligible ΔPSC_0 - the LOD would be approximately 8.5 ppm. Please justify if you believe otherwise.

Response:

We are thankful to the reviewer for the critical comments regarding the SD and slope computation. For the calculation of standard deviation (SD, before NO_2 exposure), a small number of points (~10) were usually utilized to calculate the noise factor in other studies.^{R1,R2,R3} As the reviewer suggested, we recalculated the SD considering all the points before NO_2 exposure.

Regarding slope calculation, we made an unintentional mistake, and the error in slope value further propagated to the LOD calculation. Thus, we replotted the device response (ΔPSC) as a function of NO_2 gas concentration, as shown in **Fig. R4** below.

Here we elaborately describe the method of LOD calculation:

Method 1:

- a) We executed the linear fitting of the response (Δ PSC) versus concentration plot and extracted the slope from the fitted data. After linear fitting, the slope of the fitted line finds to be 5.1355×10^{-8} A/ppm, *i.e.*, 0.05136 μ A/ppm, as shown in the *upper* panel of **Fig. R4**.
- b) Then we plotted the response (Δ PSC) as a function of time before NO₂ exposure and applied the 5th-order polynomial fitting^{R4} (see the *bottom* panel of **Fig. R4**).

Fig. R4. The *upper* panel shows the linear fitting of response versus concentration plot. The *bottom* panel shows the 5th-order polynomial fitting of the data before NO₂ exposure.

Table R1. Residual values after 5th-order polynomial fitting of the data before NO₂ exposure.

Time (s)	$(Y_i - \bar{Y})$	$(Y_i - \bar{Y})^2$	Time (s)	$(Y_i - \bar{Y})$	$(Y_i - \bar{Y})^2$
0.74653	-5.34E-09	2.85E-17	26.19743	2.78E-08	7.70E-16
1.33843	3.93E-09	1.54E-17	26.7893	-4.95E-09	2.45E-17
1.9303	2.17E-09	4.71E-18	27.38119	4.27E-10	1.82E-19
2.52218	-1.27E-08	1.62E-16	27.97307	1.92E-08	3.69E-16
3.11408	-1.65E-08	2.72E-16	28.56496	-7.40E-10	5.48E-19
3.70597	2.41E-08	5.80E-16	29.15682	-3.64E-08	1.32E-15
4.29784	1.08E-08	1.17E-16	29.74869	2.43E-08	5.89E-16
4.88973	1.44E-08	2.08E-16	30.34058	-6.19E-10	3.83E-19
5.4816	8.17E-09	6.67E-17	30.93245	-1.19E-08	1.41E-16
6.07347	3.67E-08	1.35E-15	31.5243	3.11E-09	9.69E-18
6.66538	-3.02E-08	9.13E-16	32.11617	-4.83E-09	2.33E-17
7.25726	-2.44E-08	5.95E-16	32.70806	-2.38E-08	5.67E-16
7.84914	-2.82E-08	7.95E-16	33.29995	-1.61E-08	2.60E-16
8.44103	-1.02E-08	1.04E-16	33.89183	3.03E-08	9.20E-16
9.03292	1.61E-08	2.60E-16	34.48368	1.22E-08	1.48E-16
9.62478	-1.45E-08	2.09E-16	35.07558	8.55E-09	7.31E-17
10.21664	-8.30E-09	6.89E-17	35.66746	5.50E-09	3.02E-17
10.80852	8.97E-09	8.05E-17	36.25934	-1.94E-08	3.75E-16
11.40038	3.27E-08	1.07E-15	36.85123	1.83E-08	3.34E-16
11.99228	2.15E-09	4.63E-18	37.4431	-2.58E-08	6.64E-16
12.58417	-7.49E-09	5.61E-17	38.03497	4.25E-08	1.80E-15
13.17604	-1.90E-08	3.62E-16	38.62684	-1.29E-08	1.66E-16
13.76794	-2.01E-08	4.04E-16	39.21871	3.10E-08	9.58E-16
14.35983	1.68E-08	2.81E-16	39.8106	-1.34E-08	1.81E-16
14.95171	5.87E-09	3.45E-17	40.40246	-9.67E-09	9.35E-17
15.54361	-7.57E-09	5.73E-17	40.99434	5.48E-09	3.00E-17
16.13547	1.93E-08	3.71E-16	41.58624	1.99E-08	3.95E-16
16.72734	1.89E-09	3.59E-18	42.17812	-1.38E-08	1.90E-16
17.31921	9.22E-09	8.50E-17	42.76999	-1.22E-08	1.48E-16
17.91109	1.31E-08	1.71E-16	43.3619	-1.51E-08	2.29E-16
18.50298	8.38E-09	7.03E-17	43.95378	-5.66E-09	3.21E-17
19.09486	-2.10E-08	4.39E-16	44.54564	-9.67E-09	9.35E-17
19.68674	1.69E-09	2.85E-18	45.13752	9.97E-10	9.94E-19
20.27862	-2.15E-09	4.61E-18	45.72941	-1.54E-08	2.36E-16
20.87051	-1.02E-09	1.04E-18	46.32128	2.94E-08	8.66E-16
21.46238	2.44E-08	5.96E-16	46.91319	-2.20E-08	4.85E-16
22.05427	-2.09E-08	4.35E-16	47.50507	1.37E-08	1.87E-16
22.64614	8.93E-09	7.97E-17	48.09695	-1.78E-09	3.18E-18
23.238	1.72E-08	2.95E-16	48.68884	1.16E-08	1.35E-16
23.82988	-2.10E-08	4.43E-16	49.28073	1.57E-08	2.47E-16
24.42178	-1.19E-08	1.41E-16	49.87262	-1.73E-09	3.01E-18
25.01367	-1.96E-08	3.84E-16	50.46451	-1.42E-08	2.01E-16
25.60555	-1.27E-08	1.62E-16			

- c) The SD value (1.7283×10^{-8} A, see the *bottom* panel of **Fig. R4**) is obtained by calculating the residual ($Y_i - \bar{Y}$) of the 5th-order polynomial fit (see **Table R1**) using the equation as described below:

$$SD = \sqrt{\sigma^2 / (N - 1)} \text{ where } \sigma^2 = \sum_{i=1}^N (Y_i - \bar{Y})^2$$

- d) Finally, the LOD value is calculated as follows:

$$LOD (ppm) = \frac{3 \times SD}{Slope} = \frac{3 \times 1.7283 \times 10^{-8}}{5.1355 \times 10^{-8}} = 1.01$$

As per the reviewer's suggested method, we further recalculated the LOD value by considering $\Delta PSC(0)$ and the y-intercept. Here also elaborately describe the technique of LOD calculation:

Method 2:

- a) We executed the linear fitting of the response (ΔPSC) versus concentration plot and extracted the slope (5.1355×10^{-8} A/ppm) and y-intercept (4.79508×10^{-8} A) value from the fitted data, as shown in the *upper* panel of **Fig. R5**.

Fig. R5. The *upper* panel shows the linear fitting of response versus concentration plot. The *bottom* panel shows the average value of ΔPSC before NO_2 exposure (i.e., $\Delta PSC(0)$).

Table R2. Residual values after considering the average value of Δ PSC before NO₂ exposure.

Time (s)	$(Y_i - \bar{Y})$	$(Y_i - \bar{Y})^2$	Time (s)	$(Y_i - \bar{Y})$	$(Y_i - \bar{Y})^2$
0.74653	-9.38E-08	8.80E-15	26.19743	5.31E-08	2.82E-15
1.33843	-8.29E-08	6.87E-15	26.7893	2.21E-08	4.90E-16
1.9303	-8.29E-08	6.87E-15	27.38119	2.91E-08	8.48E-16
2.52218	-9.59E-08	9.20E-15	27.97307	4.94E-08	2.44E-15
3.11408	-9.78E-08	9.56E-15	28.56496	3.08E-08	9.46E-16
3.70597	-5.52E-08	3.04E-15	29.15682	-3.70E-09	1.37E-17
4.29784	-6.63E-08	4.40E-15	29.74869	5.80E-08	3.36E-15
4.88973	-6.05E-08	3.66E-15	30.34058	3.40E-08	1.16E-15
5.4816	-6.45E-08	4.16E-15	30.93245	2.35E-08	5.54E-16
6.07347	-3.35E-08	1.12E-15	31.5243	3.91E-08	1.53E-15
6.66538	-9.80E-08	9.60E-15	32.11617	3.17E-08	1.00E-15
7.25726	-8.96E-08	8.03E-15	32.70806	1.31E-08	1.71E-16
7.84914	-9.08E-08	8.24E-15	33.29995	2.10E-08	4.40E-16
8.44103	-7.01E-08	4.91E-15	33.89183	6.75E-08	4.56E-15
9.03292	-4.10E-08	1.68E-15	34.48368	4.94E-08	2.44E-15
9.62478	-6.87E-08	4.71E-15	35.07558	4.57E-08	2.08E-15
10.21664	-5.96E-08	3.55E-15	35.66746	4.24E-08	1.80E-15
10.80852	-3.93E-08	1.55E-15	36.25934	1.73E-08	2.98E-16
11.40038	-1.26E-08	1.58E-16	36.85123	5.45E-08	2.97E-15
11.99228	-4.00E-08	1.60E-15	37.4431	1.00E-08	1.01E-16
12.58417	-4.65E-08	2.17E-15	38.03497	7.78E-08	6.05E-15
13.17604	-5.49E-08	3.02E-15	38.62684	2.19E-08	4.80E-16
13.76794	-5.28E-08	2.79E-15	39.21871	6.52E-08	4.25E-15
14.35983	-1.28E-08	1.63E-16	39.8106	2.03E-08	4.11E-16
14.95171	-2.05E-08	4.19E-16	40.40246	2.35E-08	5.54E-16
15.54361	-3.07E-08	9.43E-16	40.99434	3.82E-08	1.46E-15
16.13547	-6.77E-10	4.58E-19	41.58624	5.22E-08	2.72E-15
16.72734	-1.49E-08	2.21E-16	42.17812	1.82E-08	3.31E-16
17.31921	-4.40E-09	1.94E-17	42.76999	1.96E-08	3.83E-16
17.91109	2.58E-09	6.67E-18	43.3619	1.66E-08	2.74E-16
18.50298	9.53E-10	9.09E-19	43.95378	2.61E-08	6.81E-16
19.09486	-2.54E-08	6.43E-16	44.54564	2.24E-08	5.01E-16
19.68674	2.55E-10	6.49E-20	45.13752	3.35E-08	1.13E-15
20.27862	-6.77E-10	4.58E-19	45.72941	1.79E-08	3.22E-16
20.87051	3.28E-09	1.08E-17	46.32128	6.38E-08	4.07E-15
21.46238	3.15E-08	9.89E-16	46.91319	1.38E-08	1.89E-16
22.05427	-1.12E-08	1.24E-16	47.50507	5.12E-08	2.63E-15
22.64614	2.12E-08	4.50E-16	48.09695	3.80E-08	1.44E-15
23.238	3.19E-08	1.02E-15	48.68884	5.40E-08	2.92E-15
23.82988	-3.94E-09	1.55E-17	49.28073	6.13E-08	3.75E-15
24.42178	7.47E-09	5.58E-17	49.87262	4.75E-08	2.26E-15
25.01367	1.88E-09	3.55E-18	50.46451	3.94E-08	1.55E-15
25.60555	1.07E-08	1.15E-16			

- b) The $\Delta\text{PSC}(0)$ value (9.3809×10^{-8} A) is obtained by taking the average of ΔPSC under null NO_2 stimuli, as shown in the *bottom* panel of **Fig. R5**.
- c) The SD value (4.6709×10^{-8} A, see the *bottom* panel of **Fig. R5**) is obtained by calculating the residual ($Y_i - \bar{Y}$) of the average value of the ΔPSC under null stimuli (see **Table R2**) using the equation as described below:

$$SD = \sqrt{\frac{\sum_{i=1}^N (Y_i - \bar{Y})^2}{(N - 1)}} = \sqrt{\frac{1.8327 \times 10^{-13}}{84}} = 4.6709 \times 10^{-8}$$

- d) Finally, the LOD value is calculated with the following formula:

$$\begin{aligned} LOD \text{ (ppm)} &= \frac{\Delta\text{PSC}(0) + 3 \times SD - y\text{-intercept}}{\text{Slope}} \\ &= \frac{9.3809 \times 10^{-8} + 3 \times 4.67097 \times 10^{-8} - 4.79508 \times 10^{-8}}{5.1355 \times 10^{-8}} \\ &= 3.62 \end{aligned}$$

The LOD values obtained using both methods were found to be in a similar range (1~3 ppm) and summarized in **Table R3**.

Table R3. Calculation of SD and LOD values.

Method	Slope (A·ppm ⁻¹)	σ^2	SD	LOD (ppm)
1	5.1355×10^{-8}	2.5091×10^{-14}	1.7283×10^{-8}	1.01
2	5.1355×10^{-8}	1.8327×10^{-13}	4.6709×10^{-8}	3.62

Thus we corrected the value of LOD in both supplementary (**Supplementary Fig. S10**) and the main manuscript and employed **Method 1** to calculate the value of LOD.

Fig. S10. (a) Time-dependent measurement of the drain current (I_d) upon exposure to increasing durations of chemical pulses at a NO_2 pulsing concentration of 400 ppm. (b) Time-dependent

measurement of I_d upon exposure to increasing NO₂ concentrations in a step-like manner from 300 to 800 ppm, with a gas pulse duration of 30 s. (c) Time-dependent response (Δ PSC) of the ACNS towards the low concentration of NO₂ (10, 30, 50 and 100 ppm) with a pulse duration of 30 s. (d) Calculation of limit of detection (LOD) from the data showing the fitted responses of the device.

The theoretical LOD was calculated as: $LOD (ppm) = \frac{3 \times \text{standard deviation}}{\text{slope}} = 1.01$. Dependence of the device response (Δ PSC) on the pulse duration (20, 30, and 60 s) of (e) 10 ppm and (f) 30 ppm NO₂ pulses.

We also corrected the main manuscript (lines 299 to 300) as follows:

“Concentration-dependent changes in PSC (*i.e.*, Δ PSC) of ACNS for lower concentration NO₂ (10, 30, 50, and 100 ppm) with a fixed pulse duration of 30 s (**Supplementary Fig. S10c**) was obtained, and the limit of detection of the device was calculated as low as 1 ppm from the data (**Supplementary Fig. S10d**).”

- R1. Zhang, X. *et al.* Ultralow detection limit and ultrafast response/recovery of the H₂ gas sensor based on Pd-doped rGO/ZnO-SnO₂ from hydrothermal synthesis. *Microsystems & Nanoengineering* 2022 8:1 **8**, 1–12 (2022).
- R2. Bai, S. *et al.* rGO modified nanoplate-assembled ZnO/CdO junction for detection of NO₂. *J Hazard Mater* **394**, 121832 (2020).
- R3. Su, T. Y. *et al.* Highly sensitive, selective and stable NO₂ gas sensors with a ppb-level detection limit on 2D-platinum diselenide films. *J Mater Chem C Mater* **8**, 4851–4858 (2020).
- R4. Bag, A. *et al.* A room-temperature operable and stretchable NO₂ gas sensor composed of reduced graphene oxide anchored with MOF-derived ZnFe₂O₄ hollow octahedron. *Sens Actuators B Chem* **346**, 130463 (2021).

3. (Fig. 4) It seems from (d) that the two upper signals are in the transient regime. If possible, it would be interesting to compute the tau and the initial and final deltaPSC values by fitting an exponential equation like the one used in Fig. S5. Fig.4(f) could then be obtained letting the signals stabilize to $3 \cdot \tau$ or $4 \cdot \tau$ and evaluating all SW norm points at these same multiples of tau. Please plot all green, yellow, and red lines in Fig.4(d) and (e) up to the end of the x-axis.

Response:

We thank the reviewer for this interesting comment. We partially agree with the reviewer's opinion regarding the state of recovery for the graphs in **Fig. 4d**. According to the reviewer's observation, the two upper signals seem to be in the transient regime, which is correct (if we consider only the inhibitory electrical pulsing part). The two lower signals with small Δt reach a stable state approximately after 200 s of the two stimulations (see the *left* panel of **Fig. R6**), and a very minute change in current is further observed, whereas the two upper graphs seem to be in a transient regime and would require a long time to reach a stable state.

But herein, the purpose of the study is to observe the overall change in PSC (ΔPSC) with time depending on both the sequence and interval between past activities (i.e., excitatory chemical stimuli and inhibitory electrical pulsing). Thus we focused more on the recovery of the signal to the baseline, i.e., PSC_{in} (before the two stimulations), and the measurement was also planned accordingly. The dashed line shows all the signals reached their initial state (i.e., $\Delta PSC=0$) within 100 s after the two stimulation.

Following the reviewer's suggestion, we have calculated the decay time constant (τ varies from 10 to 40 s) by fitting only the transient signal after the two stimulations until the instances where

all the signals recover their initial state. The synaptic weight values are calculated at 4τ after the two types of stimulation and plotted in the *right* panel of **Fig. R6**. When an inhibitory electrical stimulus is applied after the excitatory chemical stimulus (**Fig. 4d**), the ions would require a specific time to reach a stable state after the first stimulation. When the duration (Δt) is small, the output after the two stimulations would be significantly influenced by the chemical stimulus due to the gradual interaction of ions and gas molecules.

Fig. R6. The changes in PSC (ΔPSC) with excitatory chemical stimuli followed by inhibitory electrical pulsing with different time intervals (*i.e.*, Δt) between the two types of stimuli. The consequent change in SW (calculated at 4τ interval after the two types of stimulation) with Δt is plotted in the *right* panel.

When Δt is large enough, however, the ion distribution will achieve a stable state, and the inhibitory electrical stimulus will slowly remove the ions from the channel layer. This explains the slower recovery of the curves at higher Δt .

Fig. R7. The changes in PSC (ΔPSC) with inhibitory electrical stimuli followed by excitatory chemical pulsing with different time intervals (*i.e.*, Δt) between the two types of stimuli. The consequent change in SW (calculated at 4τ interval after the two types of stimulation) with Δt is plotted in the *right* panel.

When an excitatory chemical stimulus is applied after the inhibitory electrical stimulus, the long-term retention effect of the excitatory stimulus facilitates the PSC to reach a stable state quickly. All the graphs in **Fig. 4e** reach a stable state soon after the two stimulations, as further

replotted in the *left* panel of **Fig. R7**. The synaptic weight values are calculated at 4τ after the chemical stimulation and plotted in the *right* panel of **Fig. R7**.

As the reviewer suggested, we evaluated our device's tau (τ) and then calculated the SW after 4τ . We incorporated the changes and replotted the data in **Fig. 4f**, as shown below (**Fig. R8**).

Fig. R8. Variation of normalized SW values with time intervals (Δt) between the two types of stimuli (excitatory followed by inhibitory and inhibitory impulses followed by an excitatory stimulation).

As the reviewer suggested, we made the changes in **Fig. 4d** and **4e**, as shown below:

Fig. 4. Excitatory and inhibitory functions of the ACNS. (a) Chemical synapses inside glomerulus acquire different action potentials from ORN and PGC and transfer the modulated postsynaptic signals to the olfactory bulb. (b) Schematic of the ACNS, mimicking the biological olfactory system and showing the excitatory and inhibitory functions. The chemical stimulus contributes to excitatory PSC by diffusion of anions, whereas the electrical stimulus results in PSC inhibition by driving cations into the channel. The net modulated PSC is collected at the source-drain electrodes. (c) The ACNS with pulsed chemical stimuli at different concentrations followed by an equal number of electrical pulses, showing potentiation first and then inhibition functions. The changes in PSC with (d) excitatory followed by inhibitory and (e) inhibitory stimuli followed by an excitatory stimulus with different time intervals between the two types of stimuli. The consequent normalized SW is calculated and presented in (f). (g) Under random excitatory and inhibitory stimuli, the ACNS shows gradual modulation in the PSC according to the time and number of applied pulses.

We also modified the main manuscript (lines 388 to 389) as follows:

“The overall normalized change in SW (calculated at 4τ interval after the two stimulations in all cases) vs. Δt (the difference in the stimulation time between the excitatory and inhibitory events) was calculated and is presented in **Fig. 4f.**”

REVIEWERS' COMMENTS

Reviewer #1 (Remarks to the Author):

Thank you for the added material and for your detailed answers to my comments.

Concerning the calculation of the LOD: as the deltaPSC signal before NO₂ exposure seems to stabilize after 30s please - in the main manuscript - report the LOD computed with method 2 and the residual values obtained from the average value of Δ PSC after the transient (approx. 30s) to immediately before exposure.

I gladly recommend the manuscript for publication once the authors address this last remark.

RESPONSE TO REFEREES (NCOMMS-22-27007B)

Reviewer #1 (Remarks to the Author):

Thank you for the added material and for your detailed answers to my comments.

Concerning the calculation of the LOD: as the ΔPSC signal before NO_2 exposure seems to stabilize after 30 s please - in the main manuscript - report the LOD computed with method 2 and the residual values obtained from the average value of ΔPSC after the transient (approx. 30s) to immediately before exposure.

I gladly recommend the manuscript for publication once the authors address this last remark.

Response:

We are happy to know that the reviewer finds that the previous response letter addressed all of the earlier concerns in detail. A point-by-point response to the additional comment regarding the recalculation of $\Delta PSC(0)$ and LOD can be found below.

We are thankful to the reviewer for this suggestion regarding the recalculation of $\Delta PSC(0)$ value and corresponding residual values, considering all the points until NO_2 exposure after the transient (~ 30 s).

We further recalculated the LOD value by considering $\Delta PSC(0)$ and the y-intercept, per the reviewer's suggested method. Here we describe the technique of the LOD calculation again in detail:

- a) We executed the linear fitting of the response (ΔPSC) versus concentration plot and extracted the slope (5.1355×10^{-8} A/ppm) and y-intercept (4.79508×10^{-8} A) value from the fitted data, as shown in the *upper* panel of **Fig. R1**.

Fig. R1. The *upper* panel shows the linear fitting of response versus concentration plot. The *bottom* panel shows the average value of ΔPSC before NO_2 exposure (*i.e.*, $\Delta\text{PSC}(0)$).

- b) The $\Delta\text{PSC}(0)$ value (1.3014×10^{-7} A) is obtained by taking the average of ΔPSC under null NO_2 stimuli after the transient (~ 30 s), as shown in the *bottom* panel of **Fig. R1**.
- c) The standard deviation (SD) value (1.8208×10^{-8} A, see the *bottom* panel of **Fig. R1**) is obtained by calculating the residual ($Y_i - \bar{Y}$) of the average value of the ΔPSC under null stimuli (*i.e.*, $\Delta\text{PSC}(0)$); see **Table R1**) using the equation as described below:

$$SD = \sqrt{\frac{\sum_{i=1}^N (Y_i - \bar{Y})^2}{(N - 1)}} = \sqrt{\frac{1.1272 \times 10^{-14}}{(35 - 1)}}$$

$$= 1.8208 \times 10^{-8}$$

Table R1. Residual values after considering the average value of ΔPSC before NO_2 exposure.

Time (s)	$(Y_i - \bar{Y})$	$(Y_i - \bar{Y})^2$	Time (s)	$(Y_i - \bar{Y})$	$(Y_i - \bar{Y})^2$
30.34058	-2.31E-09	5.36E-18	40.99434	1.88E-09	3.52E-18
30.93245	-1.28E-08	1.64E-16	41.58624	1.58E-08	2.51E-16
31.5243	2.81E-09	7.88E-18	42.17812	-1.81E-08	3.29E-16
32.11617	-4.64E-09	2.16E-17	42.76999	-1.68E-08	2.81E-16
32.70806	-2.33E-08	5.41E-16	43.3619	-1.98E-08	3.91E-16
33.29995	-1.54E-08	2.36E-16	43.95378	-1.02E-08	1.05E-16
33.89183	3.12E-08	9.74E-16	44.54564	-1.40E-08	1.95E-16
34.48368	1.31E-08	1.70E-16	45.13752	-2.78E-09	7.73E-18
35.07558	9.33E-09	8.70E-17	45.72941	-1.84E-08	3.38E-16
35.66746	6.07E-09	3.68E-17	46.32128	2.75E-08	7.56E-16
36.25934	-1.91E-08	3.64E-16	46.91319	-2.26E-08	5.09E-16
36.85123	1.82E-08	3.30E-16	47.50507	1.49E-08	2.22E-16
37.4431	-2.63E-08	6.92E-16	48.09695	1.64E-09	2.70E-18
38.03497	4.15E-08	1.72E-15	48.68884	1.77E-08	3.14E-16
38.62684	-1.44E-08	2.08E-16	49.28073	2.49E-08	6.21E-16
39.21871	2.89E-08	8.34E-16	49.87262	1.12E-08	1.25E-16
39.8106	-1.61E-08	2.58E-16	50.46451	3.04E-09	9.24E-18
40.40246	-1.28E-08	1.64E-16			

d) Finally, the LOD value is calculated with the following formula:

$$\begin{aligned} LOD (ppm) &= \frac{\Delta PSC(0) + 3 \times SD - y\text{-intercept}}{Slope} \\ &= \frac{1.3014 \times 10^{-7} + 3 \times 1.8208 \times 10^{-8} - 4.79508 \times 10^{-8}}{5.1355 \times 10^{-8}} \\ &= 2.66 \end{aligned}$$

Thus we corrected the value of LOD in the supplementary (**Supplementary Fig. S10**) and the main manuscript and employed the reviewer's suggested method to calculate the value of LOD.

Fig. S10. (a) Time-dependent measurement of the drain current (I_d) upon exposure to increasing durations of chemical pulses at a NO_2 pulsing concentration of 400 ppm. (b) Time-dependent

measurement of I_d upon exposure to increasing NO_2 concentrations in a step-like manner from 300 to 800 ppm, with a gas pulse duration of 30 s. (c) Time-dependent response (ΔPSC) of the ACNS towards the low concentration of NO_2 (10, 30, 50 and 100 ppm) with a pulse duration of 30 s. (d) The *upper* panel shows the linear fitting of response versus concentration plot. The *bottom* panel shows the average value of ΔPSC before NO_2 exposure (i.e., $\Delta\text{PSC}(0)$). Calculation of limit of detection (LOD) from the data showing the fitted responses of the device. The theoretical LOD was calculated as: $LOD (ppm) = \frac{\Delta\text{PSC}(0)+3\times SD-y\text{-intercept}}{\text{slope}} = 2.66$. Dependence of the device response (ΔPSC) on the pulse duration (20, 30, and 60 s) of (e) 10 ppm and (f) 30 ppm NO_2 pulses.

We also corrected the main manuscript (line 299) as follows:

“Concentration-dependent changes in PSC (i.e., ΔPSC) of ACNS for lower concentration NO_2 (10, 30, 50, and 100 ppm) with a fixed pulse duration of 30 s (**Supplementary Fig. S10c**) was obtained, and the limit of detection of the device was calculated as low as 2.66 ppm from the data (**Supplementary Fig. S10d**).”